# Hogwild! Inference:
# Parallel LLM Generation via Concurrent Attention

**Gleb Rodionov**[†][*]
Yandex

**Roman Garipov**[*]
HSE University
Yandex

**Alina Shutova**[*]
HSE University
Yandex

**George Yakushev**[*]
HSE University
Yandex

**Erik Schultheis**[*]
IST Austria

**Vage Egiazarian**
IST Austria

**Anton Sinitsin**
Yandex

**Denis Kuznedelev**
Yandex

**Dan Alistarh**[‡]
IST Austria

## Abstract

Large Language Models (LLMs) have demonstrated the ability to tackle increasingly complex tasks through advanced reasoning, long-form content generation, and tool use. Solving these tasks often involves long inference-time computations. In human problem solving, a common strategy to expedite work is collaboration: by dividing the problem into sub-tasks, exploring different strategies concurrently, etc. Recent research has shown that LLMs can also operate in parallel by implementing explicit cooperation frameworks, such as voting mechanisms or the explicit creation of independent sub-tasks that can be executed in parallel. However, each of these frameworks may not be suitable for all types of tasks, which can hinder their applicability. In this work, we propose a different design approach: *we run LLM "workers" in parallel, allowing them to synchronize via a concurrently-updated attention cache and prompt these workers to decide how best to collaborate.* Our approach allows the LLM instances to come up with their own collaboration strategy for the problem at hand, all the while "seeing" each other's memory in the concurrent KV cache. We implement this approach via **Hogwild! Inference:** a parallel LLM inference engine where multiple instances of the same LLM run in parallel **with the same attention cache**, with "instant" access to each other's memory.[1] Hogwild! Inference takes advantage of Rotary Position Embeddings (RoPE) to avoid recomputation while improving parallel hardware utilization. We find that modern reasoning-capable LLMs can perform inference *with shared Key-Value cache* out of the box, without additional fine-tuning.

## 1 Introduction

Many recent advancements of Large Language Models can be attributed to their ability to perform inference-time computations to improve performance [Suzgun et al., 2022, Snell et al., 2024, Beeching et al., Muennighoff et al., 2025]. This includes chain-of-thought (CoT) reasoning [Wei et al., 2022, Kojima et al., 2022, Zhang et al., 2022, Yao et al., 2023, Lightman et al., 2023], long-form generation [Bai et al., 2024] and interacting with external tools [Schick et al., 2023, Qin et al., 2023, Yao et al., 2022, Shen et al., 2023]. Popular LLM-based services have capabilities for reasoning and tool use [OpenAI et al., 2024, Google DeepMind, 2025, Anthropic, 2024]. At the same time, several reasoning-capable open-access LLMs have recently been released to the public [DeepSeek-AI et al., 2025, Qwen Team, 2025, Yang et al., 2024, Muennighoff et al., 2025, Ye et al., 2025].

Using these models to solve complex problems often requires long sequential computations, that is, generating text token-by-token. However, many reasoning problems are not sequential. Leveraging this intuition, several recent works propose parallel inference strategies that allow multiple LLMs

---

[1]Our implementation is available at `https://github.com/eqimp/hogwild_llm` .

[†]Corresponding author: `rodionovgleb@yandex-team.ru`.   [*] Equal contribution.   [‡] Senior author.

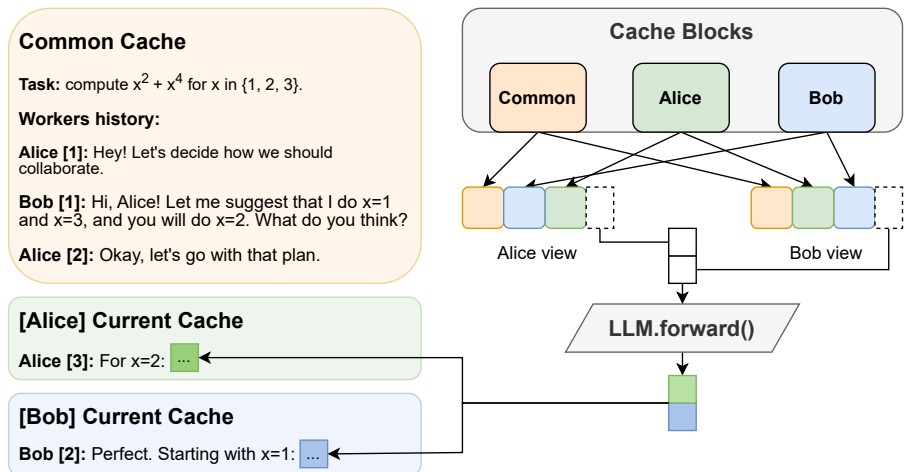

Figure 1: An intuitive explanation of Hogwild! Inference, with 2 workers generating in parallel and 3 shared cache blocks. Each color denotes a cache block. See it in action (example generation).

to solve a problem faster or more accurately via some form of collaboration [Wang et al., 2022, Ning et al., 2024]. In the simplest case, multiple LLMs can attempt the problem independently, then vote [Wang et al., 2022] or cross-reference their results [Du et al., 2023, Wang et al., 2024a] to improve correctness. A parallel line of work allows the LLM to divide the problem into multiple independent sub-tasks that are then solved in parallel and merged, producing the final solution [Ning et al., 2024, Kim et al., 2024, Jin et al., 2025]. These parallel inference strategies can improve quality and efficiency, taking advantage of parallelism in modern hardware.

Unfortunately, no single collaboration strategy is universally effective. For instance, solving a problem in independent parallel "threads" can be inefficient when one of the threads requires a longer generation than the rest, resulting in most of the agents waiting for a straggler and wasting compute [Wang et al., 2022, 2024a]. In turn, inference with independent sub-tasks only works if the problem can immediately be split into these sub-tasks. Furthermore, if one of the agents discovers that the original plan is flawed, they will be unable to re-plan [Ning et al., 2024, Ding et al., 2025], potentially solving sub-tasks that are no longer necessary [Jin et al., 2025].

This runs contrary to how humans collaborate. Instead of strict adherence to a fixed collaboration strategy, we often collaborate more dynamically, re-planning on the fly, abandoning some tasks half-way and switching to a more promising approach, discussing or debating strategy if the initial plan failed. While this type of collaboration is harder to define, it offers greater flexibility and can be more efficient if the participants are sufficiently cohesive [Hutchins, 1995, Entin and Serfaty, 1999].

**Our Approach.** In this work, we try to apply the same principle to artificial reasoners. Since modern LLMs can already reason and plan [Zhou et al., 2024, Gao et al., 2024, Wang et al., 2024c], we hypothesize that they can benefit from dynamic interaction between different instances, during which they can develop their own collaboration strategy for the problem at hand.

To test this hypothesis, we propose Hogwild! Inference — a parallel LLM inference protocol with no pre-defined framework for collaboration.[2] Instead of choosing how LLMs should interact ahead of time, we allow them to generate tokens in parallel and "see" each other's progress (tokens) **immediately as they are generated**. We then prompt the LLM "workers" to decide their next course of action by themselves, given the latest actions from others: whether this means solving parallel sub-tasks, cross-verifying each other, discussing strategy, or pivoting to a new plan.

To enable this type of on-the-fly collaboration, Hogwild! Inference runs multiple LLM instances *with the same weights*, but with a *custom Key-Value cache* that shares token representations between workers, allowing concurrent cross-attention. Specifically, instead of re-computing Key-Value representations for each worker, we keep track of individual worker KV memories and "stitch them together" in different orders, by adjusting their positional embeddings (see Figure 1). Moreover, we provide an efficient implementation of this inference approach.

We test Hogwild! Inference with modern open-source LLMs and find that existing reasoning-capable models—such as QwQ [Qwen Team, 2025] and DeepSeek-R1 [DeepSeek-AI et al., 2025]—can already "reason to coordinate". More concretely, we observe that concurrent agents can formulate and follow plans, adapt when the initial plan has failed, point out each other's errors, and use each other's

---

[2]Our approach inspired by Hogwild! SGD [Recht et al., 2011] that runs updates asynchronously and applies each update as soon as it is computed. The exclamation mark is part of the original name [Stanford HAI, 2023].

key observations. When prompted to check if they are doing redundant work – e.g., when one LLM instance is doing a sub-task that is already done by another, or solving a problem that is no longer relevant — they can often (but not always) detect redundancy and change strategy. In summary, our results suggest that parallel inference with a shared Key-Value cache may offer a promising approach to enable effective and efficient collaboration between multiple LLM instances.

## 2 Background

Recent works propose a large number of frameworks for parallel reasoning and tool use that vary across several axes: how the parallel instances are organized together, what they exchange, and how often [Zhang et al., 2025]. In this section, we give a brief summary of these methods.

**Discussion & aggregation.** The simplest way to parallelize chain-of-thought reasoning is Self-Consistency [Wang et al., 2022], where multiple LLM instances reason independently, then vote on the final answer. This approach was later extended in Du et al. [2023], replacing majority voting with text-based communication rounds. Subsequent works in this field combine multiple LLM types [Wang et al., 2024a] and scales to more agents Li et al. [2024a]. Another line of work introduces specialized "roles" such as the Debugger [Talebirad and Nadiri, 2023], Examiner [Cohen et al., 2023], Math Teacher [Kong et al., 2024], Judge [Chen et al., 2024], and others, to further augment reasoning.

This type of role-based discussion was shown to greatly improve LLM reasoning factuality for certain tasks [Wang et al., 2022, Du et al., 2023], and can even enable multiple weaker LLM agents to collectively outperform state-of-the-art single-agent systems [Wang et al., 2024a]. However, this improvement is not unique to multiple agents and can be offset with better single-agent prompting [Wang et al., 2024b, Muennighoff et al., 2025]. Additionally, these approaches do not necessarily accelerate reasoning, because at least some of the agents have to solve the entire problem sequentially, and process (re-encode) each other's progress. This creates additional computational overhead, which presents challenges for both runtime and memory efficiency Wang et al. [2024a], Du et al. [2023].

**Parallelism for efficiency.** A different line of work leverages multiple LLMs to solve tasks faster in parallel, such as Skeleton-of-Thought (SoT) [Ning et al., 2024]. SoT begins by running a single LLM to outline a plan for solving the problem with independent sub-tasks, then launches parallel LLM instances for each sub-task. For problems that involve function calling, these functions can also run in parallel [Kim et al., 2024, Gim et al., 2024]. Subsequent works propose more complex parallelism strategies such as dynamic parallel tree search [Ding et al., 2025] or a single agent spawning asynchronous sub-tasks that are done by background LLM "threads" [Jin et al., 2025, Liu et al., 2024b, Pan et al., 2025], achieved with specialized fine-tuning.

These techniques are known to substantially accelerate inference for problems that fit their type of parallelism. However, we argue that this is also their main limitation: by imposing a specific parallelism strategy, these methods can harm reasoning for problems that do not fit their framework. For instance, when solving a complex reasoning problem, it is often the case that the initial plan turns out to be wrong or incomplete [Muennighoff et al., 2025, DeepSeek-AI et al., 2025], which conflicts with SoT-like methods [Ning et al., 2024, Yu, 2025] that follow a fixed plan-execute-aggregate schedule. Furthermore, some of the sub-tasks may turn out to be more complicated than originally intended and take up more work, which would cause methods like PASTA Jin et al. [2025] to wait for that single task, whereas a more sophisticated reasoner could adjust the plan to work better in parallel. Note that each individual issue can be amended with yet another, more complicated parallelism framework, but the sheer number of such cases makes us doubt whether this is the right approach. In this work, we instead let multiple LLM instances interact without a fixed framework, allowing them to see each other's partial generations to devise (and revise) task-specific collaboration strategy. We show that, perhaps surprisingly, existing reasoning LLMs already have the ability to leverage this.

## 3 Hogwild! Inference

Our main intuition is that modern LLMs do not need a pre-defined framework for inference-time parallelism: they can organize by themselves. To test this hypothesis, we design a parallel inference protocol where multiple LLM instances can collaborate as flexibly as possible. Instead of assigning each "worker" to a specific role or sub-task, we run them together and prompt them to collaborate. This approach has two key problems: how to run multiple inference threads from the same Key-Value memory, and how to prompt LLM "workers" to collaborate over said memory. We outline how to perform LLM inference with a shared cache in Section 3.1, describe our cache structure in Section 3.2 and prompting strategy in Section 3.3. Finally, Section 3.4 describes the inference algorithm.

## 3.1 Concurrent Attention with Shared Key-Value Cache

The core ingredient of Hogwild! Inference is a shared Key-Value memory (KV cache) accessible to all workers. The cache consists of several blocks that can be reused between workers, implementing a **concurrent version of the attention mechanism** [Bahdanau et al., 2015, Vaswani, 2017].

Let us first consider a simple case with two workers and three cache blocks, as depicted in Figure 1. The first block contains the prompt, and the other two blocks contain the tokens generated by workers A and B respectively (denoted Alice and Bob in the Figure). As workers generate new tokens, they access each other's attention caches as though these were their own previously generated tokens. In Figure 1, "Alice" sees the common prompt, then "Bob's" token representations, then her own. In turn, Bob sees the same common prompt, then Alice's token KVs, and his own tokens after that.[3]

This creates a discrepancy where the same Key-Value pairs appear at different positions for each worker. Furthermore, the relative distance between the same pair of tokens (e.g., first generated tokens from Alice and Bob, respectively) changes as new tokens are added. While it is possible to re-encode these tokens at their new positions, it would cause overhead that scales cubically[4].

Instead of re-encoding the new tokens for other workers, we attempt to reuse existing token representations between workers. However, since these tokens appear at different positions for each worker and step, we need to adjust for their positional embeddings. Most modern LLMs use Rotary Position Embeddings (RoPE) [Su et al., 2021], where each key and query is rotated to an angle proportional to its absolute position. Prior works have shown that RoPE embeddings can be manipulated through scaling [Peng et al., 2023] slicing [Xiao et al., 2024], or pruning [Zhang et al., 2023].

In Hogwild! Inference, we instead shift the KV values, multiplying the entire cache block by a cos / sin values that implement rotation by a constant offset. We use this to arrange the same cache entries in different order for each worker as in Figure 1 (right). This allows both workers to instantly "see" each other's tokens while they are generated — and even before they are processed by all layers.

## 3.2 Cache Structure

Now that we defined a way to rearrange cache blocks on the fly, it is reasonable to ask how to arrange these blocks. For short tasks, simply concatenating worker outputs is sufficient. However, as we consider harder problems that require long chains of thought, workers will eventually pay less attention to each other because of the thousands of tokens between their latest steps[5].

To address this problem, we propose a more sophisticated cache arrangement inspired by group chat rooms. Namely, we split the generated text into reasoning "steps", roughly a paragraph in size. Whenever a given worker finishes a paragraph, (e.g. generates \n\n), we move its KV cache to the end of a shared chat-like history and let it generate the next paragraph at the end of that history. Note that workers still see each other's current (unfinished) paragraphs at the end of the shared history as they write them (see Figure 1). This way, workers always see each other's latest updates as recent tokens and can communicate more easily. For each worker $W_i$, we organize cache blocks as follows:

- **Common Cache:** a large KV cache block that stores KV representations for the system prompt, task description, *and a history of previous reasoning steps from each agent*.
- **Other workers:** multiple smaller cache blocks containing the latest (unfinished) steps of all other workers $W_{j \neq i}$ in ascending order. For instance, if there are 4 workers, $W_2$ will see $W_1 \oplus W_3 \oplus W_4$.
- **Current worker:** the latest (unfinished) reasoning step of the current worker $W_i$ to be continued.

Each block starts with a new paragraph (\n\n) followed by a short header text that contains worker id (Alice, Bob, etc.) and the index of the current step for the worker. Whenever a worker completes a reasoning step, their KV cache entries are moved to the end of the shared history cache block with the proper rotation, then their local cache is reset for a new step. We refer to Figure 1 for an illustration of this layout for two workers. We describe alternative (simpler) layouts in Appendix A.

## 3.3 Prompting for Zero-Shot Collaboration

The shared key-value cache inference we described above *allows* modern LLMs to access each other's tokens and reason collaboratively. However, even though modern LLMs can reason about

---

[3]For clarity of exposition, we choose to anthropomorphize the pronouns for these two LLM instances.

[4]If $n$ agents generate one new token each, which is then re-encoded differently for each of these $n$ agents, that each have to attend to $O(n)$ additional tokens, then the total step complexity is $O(n^3)$.

[5]In other words, if we put all outputs of worker A ahead of worker B, then the more tokens are generated, the farther worker B needs to "look" to reach worker A's latest outputs. This could be mitigated with finetuning.

how to collaborate, there is no guarantee that they will actually do so unprompted. As with any desired LLM behavior, it can be achieved in two ways: either by training the model to generate tokens collaboratively or by prompting it in-context. In this work, we focus on the latter approach to make Hogwild! Inference easier to generalize for new models. Our prompting consists of two parts:

1. **System prompt** describes the "rules" of the shared cache and suggests that workers collaborate. This prompt goes at the beginning of either the system or user message (if not unsupported);
2. **Inserting s1-like collaboration prompts:** every thousand generated tokens, we prompt a random worker with *"Wait, am I doing redundant work? (yes/no):"* at the beginning of their next paragraph. This strategy is meant to promote collaboration and is inspired by Muennighoff et al. [2025].

The latter s1-like prompts present a curious case. We found that LLMs fine-tuned on reasoning can often become too "focused" on what it is generating currently and fail to notice that another instance has found a mistake or solved their problem earlier. However, when asked directly, they can spot redundancy and change their approach. Overall, we found that when prompted this way, LLMs often (but not always) detect redundancies in their actions and can determine the optimal course of action.

## 3.4 Inference Matters

When generating new tokens with Hogwild! Inference, we perform a forward pass on all workers in parallel, as though they were in the same batch. Instead of each sample having its own attention cache, we allow batch elements to attend to each other's KV caches at different positions. When processing newly generated tokens, we "insert" their KV representations at the end of their respective cache blocks, then arrange these cache blocks for each worker. This way both workers can immediately attend to each other's current tokens even before they are fully processed by all layers.

This leads to the following problem: since workers combine cache blocks in different order (see Figure 1), we would need to rotate the cached KVs multiple times, one for each worker. Done naïvely, this would require rotating all past token representations at every step, which is inefficient for long contexts. Fortunately, this problem can be circumvented using a property of rotation: if both query and key are rotated by the same angle, the dot product between them will not change. **Instead of rotating all previous keys, we can rotate current token queries to an equivalent angle** (Figure 2).

Suppose that a given attention layer needs to compute attention between the current token query $q$ at position $i_q$ (denoted $\rho(q, i_q)$) and a block of keys rotated to the starting position $i_k$. Instead of rotating keys, we can rotate the query to position $i_q - i_k$ and keep the KV cache as is. If there are multiple KV blocks A, B, C (**A**lice, **B**ob, **C**ommon) that need to be rotated to positions $i_k^A, i_k^B, i_k^C$ respectively, we rotate the query $q$ multiple times for each block. Formally, we can rewrite the attention dot-product:

$$\rho(q, i_q)\Big[\rho(A, i_k^A) \oplus \rho(B, i_k^B) \oplus \rho(C, i_k^C)\Big] = \rho(q, i_q - i_k^A)A \oplus \rho(q, i_q - i_k^B)B \oplus \rho(q, i_q - i_k^C)C,$$

where $\oplus$ denotes concatenation. The r.h.s. formula only rotates the current step query, i.e. a single token per worker, as opposed to the past KV blocks that can contain thousands or millions of tokens. We use this property to design an efficient implementation of our method based on Flash-Decoding [Dao et al., 2023]. We gather each KV cache block in a contiguous memory buffer and compute attention similarly to Paged Attention [Kwon et al., 2023], where one page would correspond to one cache block and the corresponding query rotations from all workers. This way, we need only one copy of each cache block and do not need to re-rotate its entries (see Appendix B).

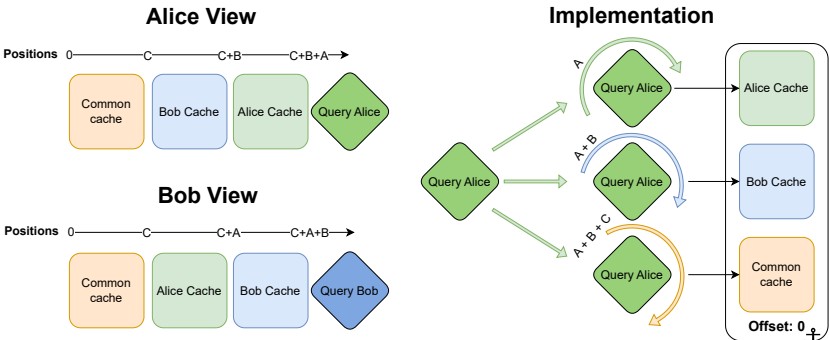

Figure 2: Intuitive scheme of Hogwild! Inference with query rotation. Colors represent cache blocks. Instead of rotating all cache blocks to align with Alice's and Bob's views, we keep them fixed at the zero position and only rotate the current token queries to equivalent angles.

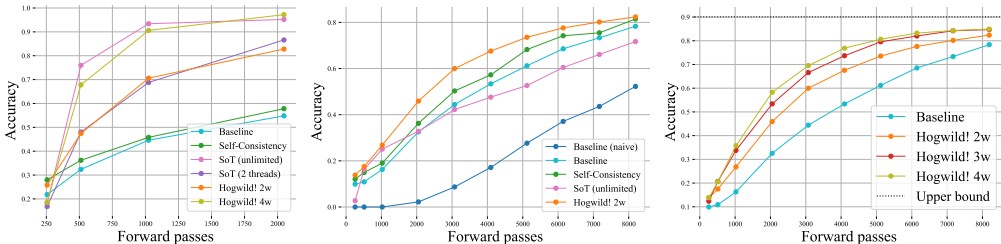

Figure 3: (left) Evaluation results for QwQ-32B on synthetic tasks with 5 GSM8k questions in each. (middle) Evaluation of Hogwild! Inference and baselines with QwQ-32B on LIMO. (right) Hogwild! Inference with varying number of workers with QwQ-32B on LIMO.

## 4 Experiments

### 4.1 Detailed Evaluation with QwQ-32B

In this section, we conduct an initial evaluation of Hogwild! Inference to test its ability to collaborate in our zero-shot setting. All evaluations in this section are done with the QwQ-32B [Qwen Team, 2025] model. We consider two tasks: one with obviously independent tasks that can be done in parallel and another with a more complicated collaboration pattern.

In both setups, we allow the model to generate reasoning up to a certain budget of sequential forward passes and evaluate its accuracy. If the model did not produce the final answer (\boxed{...}) in time, we take all generated outputs and insert a special prompt[6] that makes the model generate an answer (or its "best guess"), similarly to how it is done in Pu et al. [2025]. If there are multiple workers / threads, we feed outputs from all workers (concatenated) into the model and prompt it to generate the final answer immediately ($\leq$16 tokens, stop early if generated answer). We apply this technique to all methods except "Baseline (no early stopping)" and do not count these extra tokens towards the total budget (x axis) since they have an equal effect on all methods.

We evaluate the following generation algorithms (details in Appendix D):

- **Hogwild! Inference:** Our main algorithm, as described in Section 3. We evaluate with 2, 3 and 4 parallel "workers" and provide additional configuration details in Appendix D.1.
- **Baseline (no early stopping):** standard sequential generation with a single LLM instance. This is *the only* evaluation where we do *not* insert the early stopping prompt described above.
- **Baseline:** an improved sequential generation with the early stopping technique described above.
- **Skeleton-of-Thought (SoT) [Ning et al., 2024]:** a parallel reasoning algorithm in which the LLM first generates a short "outline" containing several independent tasks, then runs these tasks in parallel and combines the results. We run with both an unlimited number of parallel threads (original setup) and with 2 "workers" that append tokens to each thread in a round-robin fashion. For more complicated reasoning tasks, we found that Skeleton-of-Thought cannot solve the problem by itself; to mitigate this, we allow the main model to encode all generated threads and continue reasoning (with early stopping). We discuss Skeleton-of-Thought in more detail in Appendix D.2.
- **Self-consistency [Wang et al., 2022]:** a parallel reasoning algorithm where LLM instances write solutions independently, then vote on the answer. Instead of majority voting, we allow the LLM to view both solutions (concatenated) before generating the final answer with our early-stopping prompt, which outperforms voting in our setup and works even for 2 workers. Note that this method cannot split sub-tasks between workers and is instead meant to increase quality through voting.

**Sanity Checks with GSM8k$\times$5:** Before we try our approach on more challenging tasks, we test if Hogwild! Inference is capable of basic collaboration. For this purpose, we construct a toy problem set with 128 samples, each containing 5 non-overlapping questions from the GSM8k test set [Cobbe et al., 2021]. The LLM is prompted to solve each problem and return comma-separated values[7]. We report the average *per-question* accuracy, i.e. if the model solves 4 out of 5 questions in a given sample correctly, it will get a score of 0.8 for that sample.

We summarize our results in Figure 3 (left): the parallel workers under the Hogwild! Inference can indeed collaborate, i.e. our KV cache manipulations do not break down model's reasoning capabilities. As intuition suggests, Skeleton-of-Thought can also speed up this synthetic task by answering each question in parallel. We provide an example of the outline created by the Skeleton-of-Thought in Appendix E.4. Notably, the self-consistency algorithm also shows some improvement over the

---

[6]"\n\nWait, given the limited time, I have to give an answer right now. Considering all my previous attempts, I have to conclude that the final answer is \boxed{"

[7]"Solve these problems and return comma-separated answers \boxed{answer1,..., answer5} :\n 1. {task1}\n 2. {task2}\n 3. {task3}\n 4. {task4}\n 5. {task5}"

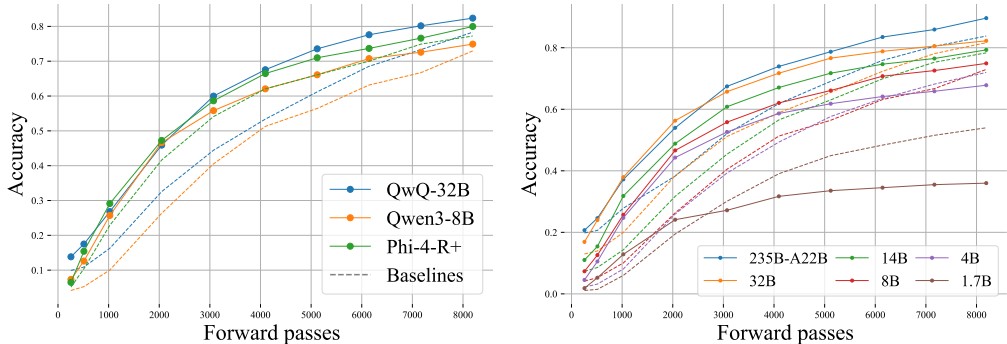

Figure 4: Evaluation of Hogwild! Inference on LIMO for QwQ-32B, Phi-4-Reasoning-Plus (14B) and Qwen3-8B (left) and different Qwen3 models (right). Dashed lines denote baselines (1 agent).

baseline, which we attribute to the fact that it gives the model two "shots" at a problem, and if one of them happens to be faster, the algorithm will on average surpass the baseline.

**LIMO tasks.** Next, we evaluate Hogwild! Inference in a more challenging setup where there is no clear pattern of collaboration. We adopt the dataset of 817 problems from Ye et al. [2025]. The dataset contains mathematical problems that take modern LLMs thousands of tokens to solve reliably. Unlike our synthetic tasks, the problems in that dataset often do not have an obvious way to agree on a collaboration strategy ahead of time, but it can emerge (and change) during reasoning.

We summarize our results in Figure 3 (middle, right). Overall, Hogwild! Inference can converge to a correct solution faster, achieving greater accuracy for the same number of consecutive steps. Furthermore, it produces greater speed-ups as we increase the number of parallel workers (though there is a limit, as we show in Appendix E.1). Similarly to our previous setup, self-consistency decoding provides some improvement over the single-worker baseline, but does not outperform Hogwild! Inference. As expected, Skeleton-of-Thought could not split the problem neatly into independent tasks, but still achieves some improvement on small budgets.

We then evaluate different LLM families and sizes on LIMO dataset in Figure 4. We found that our approach generalizes to most of the models tested, with a notable exception. For Qwen3 model family, we observe that the smaller models, 1.7B and, to a lesser extent, 4B fail to adapt to the task and get distracted from the task. In Appendix E.1, we also report additional evaluations in this setup: ablation of the cache rotation from 3.1 and our chat-like cache structure from Section 3.2. We provide examples of collaborative generations for this setup in Appendix F.

## 4.2 Additional Benchmarks and Models

Next, we test whether our approach can be generalized to other mathematical reasoning and programming tasks. For this evaluation, we also chose benchmarks that do not have obvious collaboration patterns but can nonetheless be solved faster by two human "agents". We evaluate on three such benchmarks: LiveCodeBench, OlympiadBench and AIME'25. In addition to QwQ-32B, we also report Qwen3 [Yang et al., 2025] and Phi-4 Reasoning Plus [Abdin et al., 2025]. For AIME'25, we focus on larger models and additionally include DeepSeek-R1 [DeepSeek-AI et al., 2025].

**LiveCodeBench** [Jain et al., 2024]. We evaluate on the `code_generation_lite` version `release_v5`. Our evaluation closely follows the setup from Qwen Team [2025]: we take the same 279 problems dated between 2024.08 and 2025.02 and filtered so as to avoid ones present in the QwQ dataset. Note, however, that some of the other LLMs in our setup do not report which samples, if any, did they train on. However, since we use the same model weights for the baseline and Hogwild! Inference, we can still compare the two strategies. We run the standard test suite and report Pass@1 averaged over 8 random seeds. For early stopping, we allow the method (and baseline) to generate a single final code block with up to 1024 tokens, using a similar early-stopping prompt as in Section 4.1 (see Appendix C). For Hogwild! Inference, we use the same system prompts as before.

**OlympiadBench** [He et al., 2024]. Next, we evaluate on a different reasoning benchmark that contains Olympiad-level problems on Math and Physics. We run evaluations on the two text-only english-language parts: `OE_TO_maths_en_COMP` (675 problems) and `OE_TO_physics_en_COMP` (236 problems). Unlike in Section 3, the answers to these problems are not individual numbers but LaTeX formulae that allow multiple equivalent formulations of the correct answer. We use the official evaluation codebase and adapt the built-in DeepSeek-R1 prompts for use with our model set (see details in Appendix D). For early stopping, we use the same prompt as before with 64 token limit.

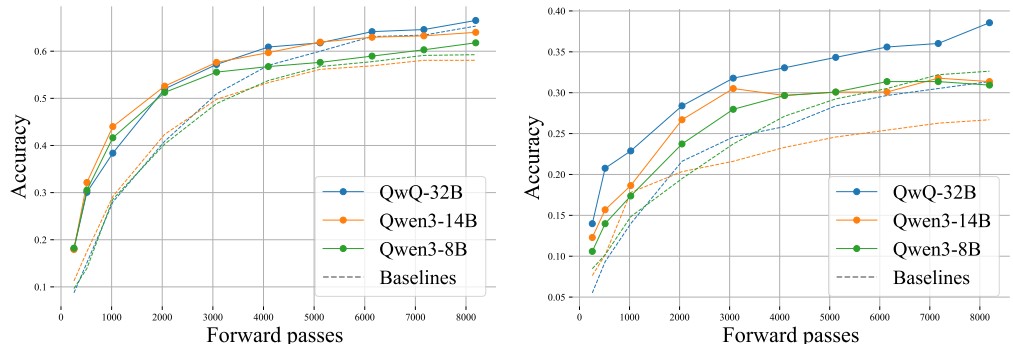

Figure 5: Evaluation of Hogwild! Inference with 2 workers on OlympiadBench Math (left) & Physics (right) for QwQ-32B, Qwen3-14B and Qwen3-8B models, dashed lines are the baselines.

**Large Models on AIME [2025].** Finally, we evaluate how Hogwild! Inference scales to larger models on a popular AIME'25 benchmark, using both I and II subsets. For this task, we focus on two models: Qwen3-235B-A22B Yang et al. [2025] and DeepSeek-R1 [DeepSeek-AI et al., 2025]. Since the AIME benchmark only contains 30 problems (15 per subset), we evaluate each model with 10 random seeds and average results. We otherwise use the same evaluation protocol as for LIMO, with the same early stopping and at most 16 tokens per answer during early stopping.

We arrange our results in Figure 5 for OlympiadBench and Figure 6 for LiveCodeBench and AIME'25. Overall, Hogwild! Inference shows similar improvements to what we observed earlier (Section 4.1). One atypical case is OlympiadBench Physics (Fig. 5 right) where Qwen3-14B stops improving after roughly 4096 tokens. Upon closer inspection, we found that the model does not break down, but overthinks the problem, improving some answers while replacing other correct answers with mistakes. Overall, the results show that the cache rotation tricks and the output structure from 3.2 can indeed be generalized across different models and benchmarks. Note, however, that due to the different output format we needed to apply slight alterations to individual model prompts: notably, QwQ-32B automatically inserts `<think>` at the end of the prompt, while Qwen3 and Phi-4 do not, so we insert it manually before the common history header. We describe this in detail in Appendix C.

## 4.3 Measuring the Ability to Collaborate

Now that we know that modern LLMs *can* collaborate in our zero-shot setting, it is natural to ask how well can they collaborate and what affects their ability. While this question deserves a more thorough investigation, we can still quantify how well LLMs collaborate under Hogwild! Inference. In this section, we analyze their "collaborativeness" using the LLM-as-a-Judge paradigm [Zheng et al., 2023a]: we feed collaborative traces into a GPT-4o [Hurst et al., 2024] model and prompt it to score behavior from 1 to 6, where "1" means no collaboration, "3" indicates basic task splitting and "6" represents a hypothetical optimal collaboration, never achieved in our analysis. We analyze LLM generations on LIMO dataset with on three models from Section 4.2. To control for differences in generation lengths we compare only 4096-token prefixes from each worker. We compare three inference setups: i) independent generations as per self-consistency decoding; ii) restricted Hogwild! Inference where agents can only view each other's finished paragraphs, but not the current (incomplete) reasoning step, and iii) full Hogwild! Inference, with 2 agents in each setup.

We summarize our scores in Figure 7: as expected, models that can see each other can collaborate and independent workers cannot. Interestingly, Hogwild! Inference with instant (token-wise) synchronization scores significantly higher than a version that can only see completed inference steps. In Appendix G we provide more detailed results, judge prompt, configurations and examples.

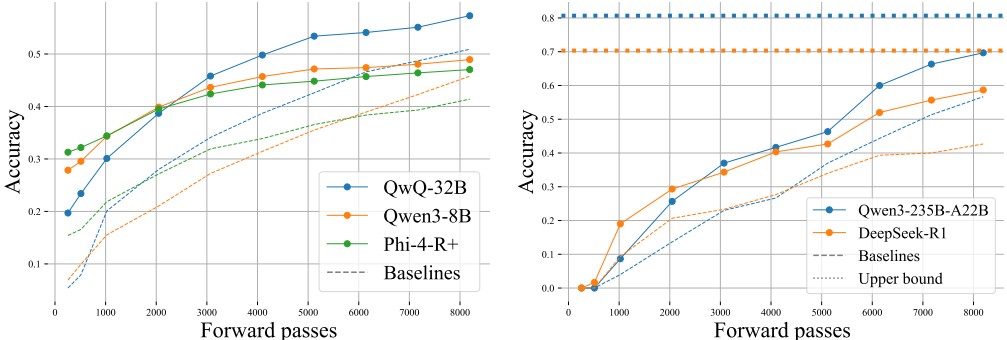

Figure 6: Evaluation of Hogwild! Inference (2 workers) on LiveCodeBench v5 2024.08-2025.02 for QwQ, Phi-4-R+ and Qwen3 (left) and AIME'25 for larger models (right), dashed lines are baselines.

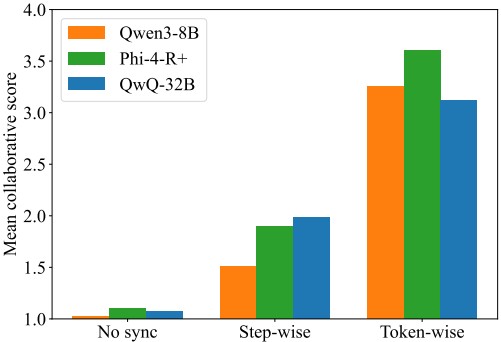

Figure 7: Mean collaborativeness score from GPT-4o. **No sync** is independent generation, **Step-wise** is restricted Hogwild! where worker can only see each-other's past steps, **Token-wise** is full Hogwild! with instant cache exchange.

Table 1: Inference benchmarks for Section 4.4. Columns denote sequence length. Rows with one worker are baselines, 2 & 4 workers use Hogwild!

| # Workers | 1024 | 2048 | 4096 | 8192 | 16384 |
|---|---|---|---|---|---|
| **Tokens per second** | | | | | |
| 1 | 20.1 | 20.0 | 19.7 | 19.3 | 18.3 |
| 2 | 36.3 | 36.2 | 36.1 | 36.1 | 34.3 |
| 4 | 68.9 | 69.0 | 69.1 | 66.3 | 60.3 |
| **Latency per forward (ms)** | | | | | |
| 1 | 49.7 | 50.0 | 50.9 | 51.7 | 54.5 |
| 2 | 55.1 | 55.3 | 55.4 | 55.3 | 58.3 |
| 4 | 58.1 | 58.0 | 57.9 | 60.4 | 66.4 |
| **Time to generate # tokens (s)** | | | | | |
| 1 | 52.3 | 103.3 | 206.5 | 416.7 | 853.5 |
| 2 | 29.9 | 58.1 | 114.6 | 228.0 | 454.4 |
| 4 | 16.7 | 31.6 | 61.3 | 120.7 | 239.2 |

## 4.4 Inference

To recall, our main motivation for proposing Hogwild! Inference is to enable faster reasoning through collaboration. Since the actual inference speed depends on many factors (GPU(s), software, precision, etc), we previously focused on evaluating inference speed in terms of the number of consecutive forward passes and not inference time. Here, in turn, we report the actual inference speed in terms of latency and tokens per second. We evaluate three setups: baseline sequential inference and Hogwild! Inference for two and four workers. We run baseline with FlashAttention v2 (FlashDecoding) and our algorithm with custom GPU kernels using the approach described in Section 3.4. We use a NVIDIA L40S GPU and AMD EPYC 9534 and benchmark the official quantized version of QwQ-32B-AWQ for all setups.

Our results in Table 1 show that, for the 32B model, Hogwild! Inference can generate tokens nearly twice as fast for 2 workers and about $3.2-3.6\times$ faster for 4 workers, which means that the accuracy gains from earlier sections can translate to faster solutions. We also report the average over GPUs, as well the 10% and 90% percentiles, in Figure 8 (left). Overall, Hogwild! Inference has a small constant latency offset compared to the baseline and near-linear scaling as we increase the number of workers. While our implementation already shows significant performance gains, we discuss several ways to scale it further in Appendix B, including in distributed setting.

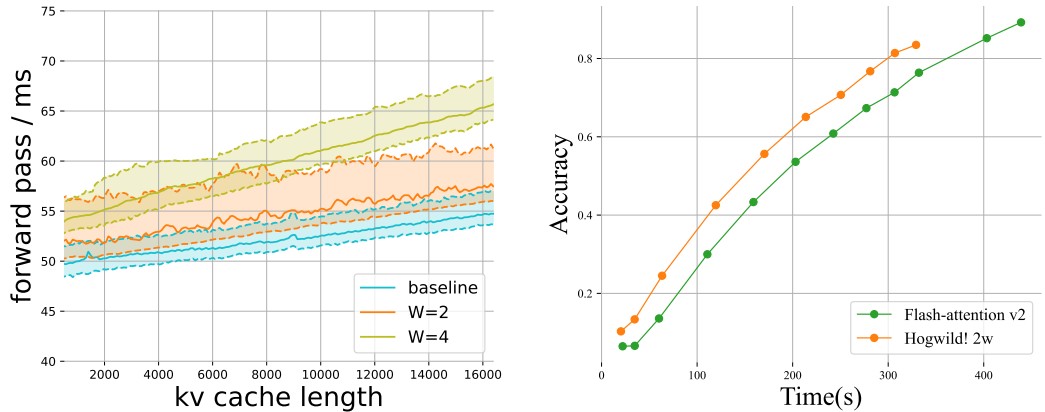

Figure 8: (left) Duration of a single forward pass (generating $W$ new tokens) for `Qwen/QwQ-32B-AWQ` on L40S, given the total number of tokens already in the KV cache. The dotted lines indicate the 10% and 90% quantiles over multiple repetitions on different GPUs. (right) Accuracy versus average generation time on the LIMO dataset task using QwQ-32B-AWQ under different token budgets.

As the figure shows, there is some overhead associated with preparing multiple caches (i.e., even at an empty cache, Hogwild! is slightly slower than pure FlashAttention). A more detailed breakdown is presented in Table 2, which shows the duration of the attention kernel (or attention+rope for Hogwild!), as well as the total setup time, that is, the time spent preparing the data structures needed for Hogwild! The latter needs to be done only once per forward pass, instead of once per transformer

Table 2: Breakdown of Hogwild! overhead compared to pure FlashAttention inference.

| KV Length | Attention (×64) | | | Setup (×1) | | |
|---|---|---|---|---|---|---|
| | FA | W2 | W4 | FA | W2 | W4 |
| 300 | 11μs | 45μs | 45μs | – | 1.9ms | 3.9ms |
| 4096 | 35μs | 65μs | 82μs | – | 1.9ms | 3.9ms |
| 8192 | 55μs | 92μs | 123μs | – | 1.9ms | 3.9ms |
| 16384 | 100μs | 140μs | 203μs | – | 1.9ms | 3.9ms |

block. For long contexts, the attention call is about 40% and 100% slower for generating with 2 and 4 workers, respectively.

Additionally, we report accuracy results over time using our kernel on the official quantized version of QwQ-32B-AWQ on LIMO dataset. The experiments were conducted on NVIDIA L40S GPUs. For comparison, we run the baseline (FlashAttention v2) and Hogwild with 2 workers, maintaining the same experimental setup as detailed in Section 4.1. We report our results in Figure 8 (right). As illustrated, our method achieves better accuracy results on the LIMO dataset within the same time budget.

## 5 Discussion

In this work, we investigated the ability of large language models to perform parallel generation where multiple instances synchronize through a shared, dynamically-updated attention cache. Surprisingly, our results show that LLMs can operate effectively in parallel across dynamically updated attention cache without specialized fine-tuning. We demonstrate that parallel inference threads can explicitly coordinate, leveraging each other's partial solutions to enable collaborative problem-solving.

The proposed method, called Hogwild! Inference, allows multiple inference threads to concurrently access and update a shared attention cache. By leveraging Rotary Position Embeddings (RoPE), our approach introduces minimal computational overhead while ensuring instant synchronization—newly generated KV cache entries becoming immediately visible to all threads. This "telepathic" communication opens up new possibilities for efficient parallel generation with LLMs.

**Limitations** Our method exhibits reduced robustness when applied to smaller models or longer contexts, suggesting scalability challenges across model sizes and sequence lengths. Additionally, our automatic evaluation metric relies on a proprietary model, which may limit reproducibility.

**Future work** In future work, we plan to investigate methods for improving collaboration between threads, such as fine-tuning and reinforcement learning. We also plan to investigate connections to alternative parallel inference schemes, such as speculative decoding [Leviathan et al., 2023], and parallel token generation methods like Medusa [Cai et al., 2024] or EAGLE [Li et al., 2024b]. Finally, it is interesting to consider alternative shared memory structures: allowing workers to insert new steps in any order, selectively delete (forget) steps, or solving programming and tool use tasks with a shared IDE and file-system. The KV cache rearrangement used in Hogwild! Inference could also allow humans to interact with agents asynchronously, giving clarifications and feedback during reasoning.

**Acknowledgements:** We thank Vladimir Malinovskii for his help with brainstorming, helpful feedback and suggesting future work directions. We also thank Philip Zmushko for proofreading.

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

# A    Cache Layouts

In this section, we consider three cache arrangements, shown at Figure 9, with progressively more complex structure.

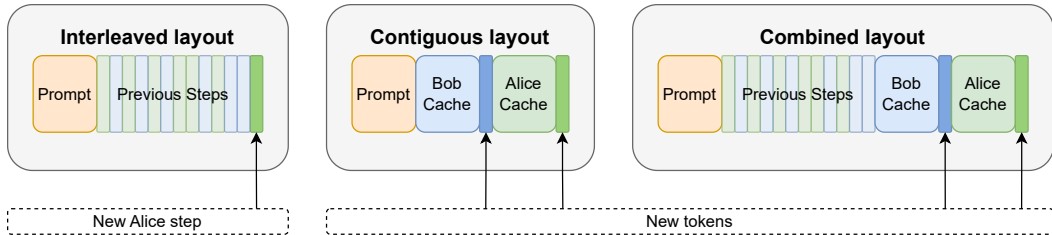

Figure 9: Three cache layouts described in Section 3.2: interleaved with step-wise synchrony (left), simple contiguous layout (middle) and combined with token-wise synchrony (right). All layouts are made from Alice point of view.

**Contiguous layout (token-wise)** is the simplest possible layout where each worker appends to their own sequence blob of tokens and sees other workers' token representations as past keys and values. This layout is inspired by collaborative text editors such as Google Docs or Overleaf.

As described earlier in Section 3.1, each worker arranges the other workers' thoughts in a different order. They see the common prompt cache first, then the caches of all *other* workers (excluding themselves[8]), then their own cache as immediate previous tokens. That way, each worker predicts the next token for their own cache.

**Interleaved layout (step-wise),** which can be seen as analogous to group chat services such as Slack or Discord. In this layout, workers generate tokens *in private* until they finish a reasoning step[9], then add it to a shared "history". The history contains past reasoning steps of each LLM instance in the order of their completion. Whenever a worker completes a reasoning step, their KV cache entries are moved to the end of the shared history cache block with the proper rotation, then their local cache is reset their local cache for a new step.

In this setup, the workers only see each other's outputs in full steps, not after every token. However, they do not wait for each other to complete their steps. Instead, each worker keeps generating new tokens and occasionally receives additional key-value pairs inserted into its cache.

**Combined layout (token-wise)** is a mixture of the first two, and is the main layout used in the paper. The LLM instances generate steps that are accumulated in a shared history, as in the interleaved layout. However, they do not generate these steps in private, but can instantly see each other's current progress, as in the contiguous layout.

We can view the first two layouts as ablated versions of this combined one: the contiguous layout lacks the shared history, and the interleaved layout lacks immediate synchronization. We compare these three layouts empirically in Appendix E.1 to better quantify the effect of each design choice.

# B    Implementation Details

Here we discuss additional implementation details and possible alternatives. To recall Section 3.4, Hogwild! inference can be implemented as a standard batched inference with a special KV "cache" that facilitates cross-worker attention.

**Cache blocks.** The Hogwild! cache is split into blocks, typically one block for each worker and an additional "common" block for prompt and past steps. The blocks contain key-value pairs for all model layers, but since all layers are treated equally, we describe the cache behavior for a single layer.

---

[8]When extending this layout to more than 2 workers, each worker sees the key-value memories of everyone except themselves. For instance, given 3 workers A, B, and C, worker B will see a version of the cache that contains the prompt, outputs of workers A and C, and finally, B's own memory. Likewise, A sees B & C, then A.

[9]We define a reasoning step as any amount of text that ends with a complete sentence, e.g. a dot or a question mark, and then a double newline (`"\n\n"`) in all our experiments, though it may vary by the model.

Within each cache block, attention keys and values are stored as though they were at positions 0, 1, ..., `len(block)`, regardless of the block's actual position in the full cache. During inference, we account for actual positions by rotating attention *queries* to the relative difference in positions (as described in Section 3.4).

**Adding new tokens to the cache.** During attention forward pass, the first thing that we do is encode the new tokens for each worker and append their keys and values to the respective cache blocks. When using RoPE, the keys are rotated not to their actual positions, but to their index within their cache block (e.g. Alice's tokens). During one inference step, these indices will be equal across all model layers — we can compute the RoPE $\sin$ and $\cos$ tensors once and reuse them between layers.

**Rotating queries.** Unlike in traditional attention, Hogwild! inference rotates query vectors multiple times for each block. Before forward pass, we calculate the difference in positions between each worker's new token (from that worker's point of view) and the first token in each KV cache block. In our main inference scenario, all $n$ workers are allowed to view each other's cache blocks plus an additional bock for prompt and history, for a total of $n \cdot (n+1)$ query rotations with exactly $n$ queries for each block. These relative positions are also equal across all layers, so we can reuse the $\sin$ and $\cos$ tensors similarly to how they are reused for keys. Note that the number of query rotations for all-to-all attention is quadratic in $n$, but it does not increase the overall time complexity of attention dot product, which is already quadratic in the number of tokens, which is always greater than $n$.

**Attention kernel.** Once we have all query rotations, we can calculate the scaled dot-product attention as usual. As our cache is naturally partitioned into smaller segments as described above, Hogwild! attention is similar to paged attention, except that each page (i.e., cache block) uses a differently rotated version of the query. A significant challenge for efficient attention in the inference setup is that for optimal data reuse, one would want to handle each KV head inside a single streaming multiprocessor (SM), so that the KV cache needs to be loaded exactly once. However, this would leave large parts of the GPU unused, as the number of KV heads can be much lower than the number of SMs. Therefore, one has to employ a form of sequnce parallelism within a single GPU, in which different SMs handle a subset of the sequence for one KV head, and a second phase handles the (cheap) reduction over partial results. Such a split-k type computation is implemented, for example, in Flash-Decoding [Dao et al., 2023].

Even though the different cache blocks used in Hogwild! would appear to be convenient points to split work across SMs, in a typical inference scenario, this would lead to very imbalanced workloads. Thus, we do not split based on cache blocks, and instead assign each SM the same number of KV entries.

**Fine-tuning and re-encoding considerations.** While our work mainly focuses on inference, fine-tuning models to perform Hogwild! inference is an interesting engineering problem. From the computational point of view, the main difference between LLM inference and fine-tuning is that inference is sequential, whereas fine-tuning can compute all positions in parallel. To fine-tune in our setup, one would want to replicate the attention computations from consecutive inference steps.

To achieve this, we record the position differences between queries and each respective cache block from each of $t$ inference steps, and how many tokens were in each block during that query, for a total of $2 \cdot t \cdot n \cdot (n+1)$ integers (negligible compared to model parameters and activations). Recall that the cache blocks always store keys and values at positions 0, 1, . . . , `len(block)`. During forward pass, these positions can be used to construct a 4D attention mask[10] to compute attention for all steps in parallel. The backward pass also runs in parallel with PyTorch autograd [Paszke et al., 2019]. A recent work by Zheng et al. [2025] explores finetuning for parallel inference in more detail.

In addition to fine-tuning, this technique can putentially be used during inference to restore generation after it was evicted from an inference server, e.g. due to preemption or hardware error mid decoding. It can also be used to re-encode in-context learning examples if they use Hogwild! inference.

**Attention variants.** Some of the recently introduced LLMs use attention variants such as Local (windowed) Attention [Rae and Razavi, 2020, Beltagy et al., 2020] or Multihead Latent Attention (MLA) [Liu et al., 2024a]. These attention variants can also be adapted for use with Hogwild! inference with minor code modifications. For local attention, queries can "skip" blocks that are outside their local window. Similarly for MLA, we can calculate compressed latent vectors within each cache block and adapt the existing MLA code to accumulate attention weights across blocks.

---

[10]`https://huggingface.co/blog/poedator/4d-masks`

**Distributed Inference.** Likewise, Hogwild! inference can be used in distributed setup using the same strategies that work for traditional attention [Shoeybi et al., 2019, Aminabadi et al., 2022]. For pipeline parallelism, each device stores cache blocks for its local subset of model layers. Likewise, for tensor parallelism, each device stores past keys of all cache blocks and layers, but only for a subset of attention heads within each layer and inference using existing kernels.

In principle, Hogwild! inference can also be combined with sequence parallelism [Jacobs et al., 2023, Liu et al., 2023], where each device stores a KV cache for a subset of tokens. One intuitive way to partition KV cache between GPUs is to assign each device to run one or several "workers" and keep the KVs generated by these workers. Since Hogwild! workers generate tokens at the same rate, each device will store the same amount of KVs and query other devices work cross-worker attention.

When computing Hogwild! concurrent attention with sequence parallelism, workers can exchange rotated queries using the All-to-All collective operation (Scatter/Gather) available in most frameworks [Li et al., 2020]. After that, each worker computes dot-products between the rotated queries and its local KV cache, and exchanges the partial results as in Ring Attention [Liu et al., 2023]. Note, however, that maximizing the performance of such sequence-parallel Hogwild! inference would require custom kernels that overlap computation and communication. In contract, tensor-parallel (per-head) an pipeline-parallel (per-layer) partitioning can reuse single-GPU attention kernels.

**Additional considerations.** Conceptually, our approach is related to the recently introduced Paged Attention from vLLM [Kwon et al., 2023] and Radix Attention from SGLang [Zheng et al., 2023b]. These techniques are similar to ours in that they perform attention to slices of all tokens, e.g. when facilitating efficient parallel beam search inference, different hypotheses attend to different (but overlapping) subsets of the KV cache. However, unlike Radix Attention, our procedure attends to all segments at once (with different rotations) and aggregates results in the same softmax-weighted sum.

## C  Prompting and formatting details

In this section, we describe the prompting and formatting details of our approach.

```
# Collaborative Reasoning
You will collaborate on this problem with another assistant.  You will write
your thoughts simultaneously with them and collaborate without redundant work.
You can collaborate by doing different parts of the problem, double-checking
each other's results, trying different approaches, or any other means.
There are 2 assistants, including yourself.  You will refer to each other as
Alice and Bob.
You will solve the problem together, writing your thoughts in parallel.  You
will be able to see each other's past and current thoughts as we write them.
You will see each other's previous steps as
**AssistantName [step]:** <...> .
In the '### Past steps' section, the automated system will gather the
thoughts of Alice and Bob as you write them.
After the '### Work in progress (others)' section, you will see the other
assistants' unfinished steps.  They will write those steps concurrently with
you.  You will take into account what they are doing.  If another assistant
gives you suggestions, you should address them.
You will always see *other* assistants' incomplete thoughts first, and
then, after '### Work in progress (own)', your own current step.  Other
assistants will continue writing their thoughts in the background while you
will continue writing your own.
Since you and others both write your thoughts in parallel, you will initially
see only partial (unfinished) thoughts that others will continue in parallel,
while you write yours.  Others' thoughts will appear at the end of their
unfinished step, near <...>.  Other assistants may write new thoughts while
you are writing yours.
You will use these partial thoughts to decide how best to collaborate without
doing the same work twice.  You will periodically check what other assistants
are doing and you should adjust your actions based on what they are doing so
you collaborate efficiently with them.
```

Prompt for collaborative inference with two workers

```
If what you are currently doing is the same thing that another assistant
has already done or is in process of doing, you will stop (e.g.  Alice may
say 'Wait, I was doing the same as Bob ...')  and change to a different task
right away, so as to avoid doing redundant work.
# Solve the following problem
Alice and Bob, you will now solve the next problem together.  Keep track of
who does what work and communicate to avoid doing the same work twice.
```

First, we provide a full prompt for collaborative reasoning involving two workers. This prompt is wrapped with standard chat template for each model. Then, all worker steps are generated in a single assistant turn. Additionally, we ensure that for reasoning models assistant turn begins with a `<think>` token - applied automatically for QwQ-32B and manually for other reasoning models. For further implementation details, we refer to the source code.

The second part of our prompting approach involves s1-like interventions [Muennighoff et al., 2025] in the generation process, where we ask whether a worker is performing redundant tasks (e.g., overlapping with another worker and can pivot to explore alternative ideas). We insert the prompt `"Quick check:  am I doing redundant work?  (yes/no):  "` at the beginning of each new reasoning step every 1024 tokens generated. We refer to Appendix F for examples that demonstrate how these interventions affect the generation process.

Next, an important part of our approach is defining the end of a reasoning step, which is needed to organize cache layout, as discussed in the paper. We define an end of step as the generation of a token containing the separator sequence (\n\n) that directly follows a token ending with an end-of-sentence marker (., ?, or !, etc). This termination condition is not met when: i) The separator appears within a generated code block (steps continue until the model completes the entire code block); ii) The preceding token ends with non-terminal punctuation (e.g., comma, colon, or semicolon)

Finally, the last part of our prompting approach is the early finisher, which allows us to extract an answer from partial reasoning chains. If the model did not produce the final answer (\boxed{...}) in time, we take all generated outputs and insert a special prompt that makes the model generate an answer (or its "best guess"), similarly to how it is done in Pu et al. [2025].

> **Prompt for early stopping**
>
> ```
> \n\nWait, given the limited time, I have to give an answer right now.
> Conside- ring all my previous attempts, I have to conclude that the final
> answer is \boxed{
> ```

After this prompt, we allow the model to generate a fixed number of tokens: 16 for LIMO and AIME, 64 for OlympiadBench, and 1024 for LiveCodeBench.

Note, however, that the LLM does not always produce the answer in time, especially with a tight budget. With QwQ-32B, we observe that the model almost always returns answers correctly if they are present, and if not, it guesses or refuses to answer (unknown, n/a or similar). When extracting answers from Hogwild! Inference, we let the final model view all generated tokens from each worker. This is equivalent to viewing the problem from the perspective of the last worker, e.g. Bob if there are two.

## D   Detailed Experiment Configuration

### D.1   Hogwild! Configuration

For the main experiments, we use Hogwild! inference with two workers (Alice and Bob), a combined layout, and the prompting techniques described in Appendix C.

### D.2   Baselines Configuration

To evaluate Skeleton-of-Thought (SoT) on our synthetic setup with grouped tasks from GSM8k, we adopt the original prompts from the paper with minor modifications. Specifically, we adjust

the prompts to ensure the model returns the answer to each subtask enclosed within \boxed{} for structured parsing.

---

**Outline prompt for Skeleton-of-Thought**

```
You're an organizer responsible for only giving the skeleton (not the full
content) for answering the question.  Provide the skeleton in a list of
points (numbered 1., 2., 3., etc.)  to answer the question.  Instead of
writing a full sentence, each skeleton point should be very short with only
3 5 words.  Generally, the skeleton should have 3 10 points.
Question:
What are the typical types of Chinese dishes?
Skeleton:
1.  Dumplings.
2.  Noodles.
3.  Dim Sum.
4.  Hot Pot.
5.  Wonton.
6.  Ma Po Tofu.
7.  Char Siu.
8.  Fried Rice.
Question:
What are some practical tips for individuals to reduce their carbon
emissions?
Skeleton:
1.  Energy conservation.
2.  Efficient transportation.
3.  Home energy efficiency.
4.  Reduce water consumption.
5.  Sustainable diet.
6.  Sustainable travel.
Now, please provide the skeleton for the following question.
{request}
Skeleton:
 [ROLESWITCHING assistant:]  1.
```

---

**Point prompt for Skeleton-of-Thought**

```
You're responsible for continuing the writing of one and only one point in
the overall answer to the following question.
{request}
The skeleton of the answer is
{outline}
Continue and only continue the writing of point {point}.  Do not continue
with other points!  Reason step-by-step and put your final answer within
\boxed{} this is very important![ROLESWITCHING assistant:]{point}.
{point_outline}
```

---

## D.3   Datasets and Benchmarks

This subsection provides links to all datasets and benchmarks referenced in this work, along with their respective licenses.

- **GSM8K**
  https://huggingface.co/datasets/openai/gsm8k
  License: MIT

- **LIMO**
  https://huggingface.co/datasets/GAIR/LIMO
  License: Apache 2.0

- **OlympiadBench**
  `https://huggingface.co/datasets/Hothan/OlympiadBench`
  License: Apache 2.0

- **LiveCodeBench**
  `https://huggingface.co/datasets/livecodebench/code_generation_lite`
  License: cc

- **AIME25**
  `https://huggingface.co/datasets/math-ai/aime25`
  License: Apache 2.0

## D.4   Compute Resources

As our approach is training-free, all computational resources were solely utilized for inference. The experiments were conducted primarily on NVIDIA A100 GPUs servers with NVSwitch, with DeepSeek-R1 experiments running in a distributed setup. The one exception to this is the inference time experiments in Section 4.4 that were run on NVIDIA L40S GPU.

The runtime per individual experiment varies by model size, benchmark and the number of workers: baseline inference with Qwen3-4B runs on LIMO in 14 hours on a single server (112 gpu-hours), whereas Qwen3-235B-A22 Hogwild! Inference ran on 40 servers for approximately 25 hours ($\approx 8K$ GPU hours). Overall, we estimate that the total GPU resources expended for this work, including early experiments that are not reported in this paper, amount to approximately $\approx 25.3K$ GPU days. Note, however, that this is largely due to the fact that we used a non-optimized inference code for most of the experimentation: the non-optimized code was developed first and we ran most of the experiments in parallel with developing the optimized version. This also means that most of our experiments under-utilized the GPUs and ran at lower power (for the purpose of environmental impact). Over 2/3 of our compute was spent on large models (Qwen3-235B-A22B and DeepSeek-R1) that utilized gpu to less than 20% (as per volatile GPU utilization) due to the use of naive model parallelism and network bottlenecks. We anticipate that future experiments can be run at significantly better utulization using the efficient implementation described in Appendix B and included in the supplementary code.

# E   Additional Experiments

## E.1   Ablation Analysis

In this section, we ablate the main components of our approach, including layouts and prompting. We use the same experimental configuration as in Sections 4.1 and 4.2 for LIMO.

In Figure 10 (left), we compare the three Hogwild! cache layouts described in Appendix A. Namely, the **Hogwild! (contiguous)** corresponds to using the contiguous cache layout where all tokens generated by a given worker are kept together, without splitting into individual steps. In turn, **Hogwild! (non-instant)** corresponds to the interleaved cache layout where workers can only see each other's past reasoning steps, but not the latest unfinished paragraph. We also ablate the use of the collaboration prompt from Section 3.3 ("Wait, am I doing redundant work?").

Finally, we test a version of Hogwild! Inference where we re-encode worker tokens instead of rotating them to a new position when moving between worker caches and the common "chat history" cache. This ablation is needed to test if our cache rotation from Section 3.1 and 3.4 is indeed an acceptable substitute for encoding tokens directly at each position (which would cause additional computational overhead). Note that, while token re-encoding is more "fair" from the perspective of position encodings, it also has a downside that it does not allow the re-encoded tokens to see some of the concurrently generated tokens from the other worker. For instance, suppose that Alice and Bob are writing steps concurrently and communicating with each other within these steps, e.g. using each other's results. Then, if we later re-encode these steps in some sequential order, then the tokens of the first worker will be encoded without access to the other worker's tokens (if it hasn't finished its

own step yet). If workers reused information from each other's steps, re-encoding this way can break some of the internal representations.

Our results suggest that all three design choices contribute to the method performance: the contiguous layout performs nearly equally well for shorter budgets, but eventually falls behind as we consider longer reasoning traces. Likewise, the interleaved layout without instant synchronization performs poorly at smaller budgets, but catches up eventually: we attribute this to the fact that slower synchronization increases the difficulty of cross-worker coordination (this also aligns with our findings in Section 4.3). The use of collaboration prompts also improves the accuracy to budget trade-offs, although we hypothesize that it can be made redundant if the model is trained to collaborate better.

In Figure 10 (right), we also compare different numbers of workers and test Hogwild! Inference with only a single worker for ablation. The results with a single worker generally perform similar to the baseline, with slightly worse accuracy for smaller budgets, which sugests that the improvements from Hogwild! Inference come from multiple workers and not as an indirect effect of our prompt. As for multiple workers, we find that using 3 and 4 workers further improves the accuracy to budget trade-offs. Curiously, as we switch to 6 workers, Hogwild! Inference performs better yet at smaller budgets, but eventually saturates at a somewhat worse accuracy.

We hypothesize that the drop of accuracy is caused by the fact that QwQ-32B was trained on a limited sequence length and, since 6 workers generate tokens at a quicker rate, the model eventually runs out of the designed maximum sequence length and performs unstably (we did not use YaRN[Peng et al., 2023] for this evaluation). However, it is also possible to attribute this to fundamental property of LIMO tasks, model limitations, our zero-shot prompt not scaling well. We leave further exploration of scaling Hogwild! Inference to multiple workers to future work.

## E.2 Detailed Model Evaluations

Due to space limitations, we had to arrange our results in Section 4.2 with multiple models per plot and had to omit some results. In this section, we report the missing evaluations on a per-model basis. In Figures 11, 12, 13, 14, 15, 16, 17, 18 we report results for QwQ, Phi-4-reasoning-plus and the Qwen3 model family. We also report limited evaluations for Llama 3.3 70B Instruct and DeepSeek-R1 in Figure 19. All evaluations are performed in the same setup as in Section 4.2.

Overall, the results align with our findings summarized in Section 4.2. Zero-shot Hogwild! Inference seems to perform better with larger models, but can be unstable for smaller ones, especially 1.7B (See Figure 13). While it is tempting to conclude that larger and more capable models are better at collaborating, it does not immediately follow from our results and can be due to some other factor. Note also that, while we observe better results with larger models, smaller Qwen3-4B and 8B models already show some signs of collaborativeness, which should make it possible to reproduce and build on our results with consumer hardware. Additionally, we hypothecize that the poor performance of 1.7B models could potentially be alleviated with finetuning in collaborative inference setup (we discuss some finetuning details in Appendix B), but we leave this to future work.

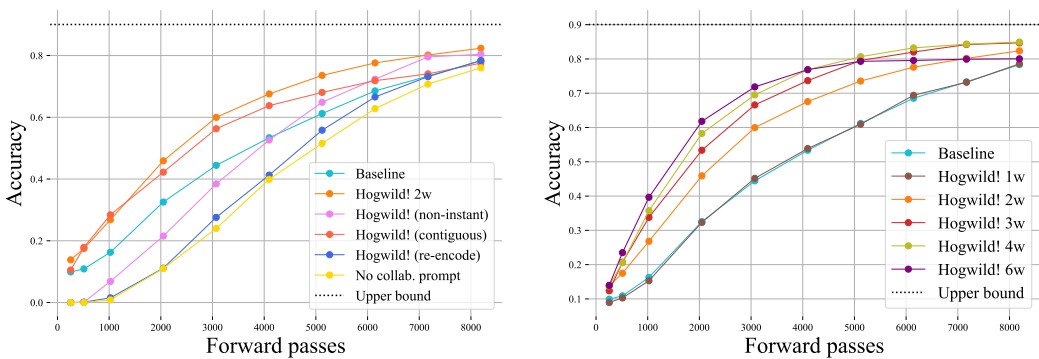

Figure 10: Detailed comparison of various parallel inference setups with QwQ-32B on LIMO task set, in the same setup as in Section 4. (left) ablation analysis of simpler cache layouts and collaboration prompt (see Section 3.3, Appendix C). (right) Hogwild! Inference with 1, 2, 3, 4 and 6 workers.

Curiously, we found that LiveCodeBench with Self-Consistency Chain-of-Thought inference [Wang et al., 2022] has significant gain in performance over the baseline. Upon closer examination, we found that the reason for this is that we always allow the model to generate a lot (up to 1024) of additional "free" tokens at the end of two generations, whereas for Hogwild! and Baseline we only generate these tokens if the model failed to produce *any* answer. If we allow Hogwild! to also generate the extra 1024 tokens all the time, its advantage also increases. However, we still report the weaker version of Hogwild! Inference and Baseline to better match our evaluation protocol on other tasks.

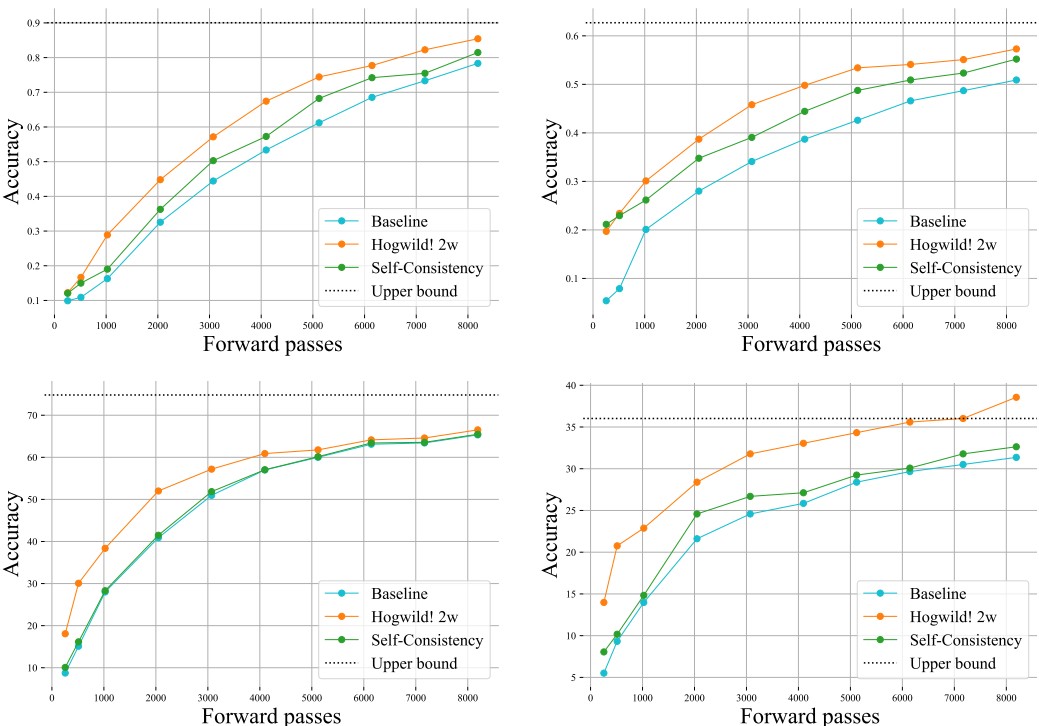

Figure 11: Results for QwQ-32B on LIMO (top-left), LiveCodeBench (top-right), OlympiadBench-Math (bottom-left) and OlympiadBench-Physics (bottom-right).

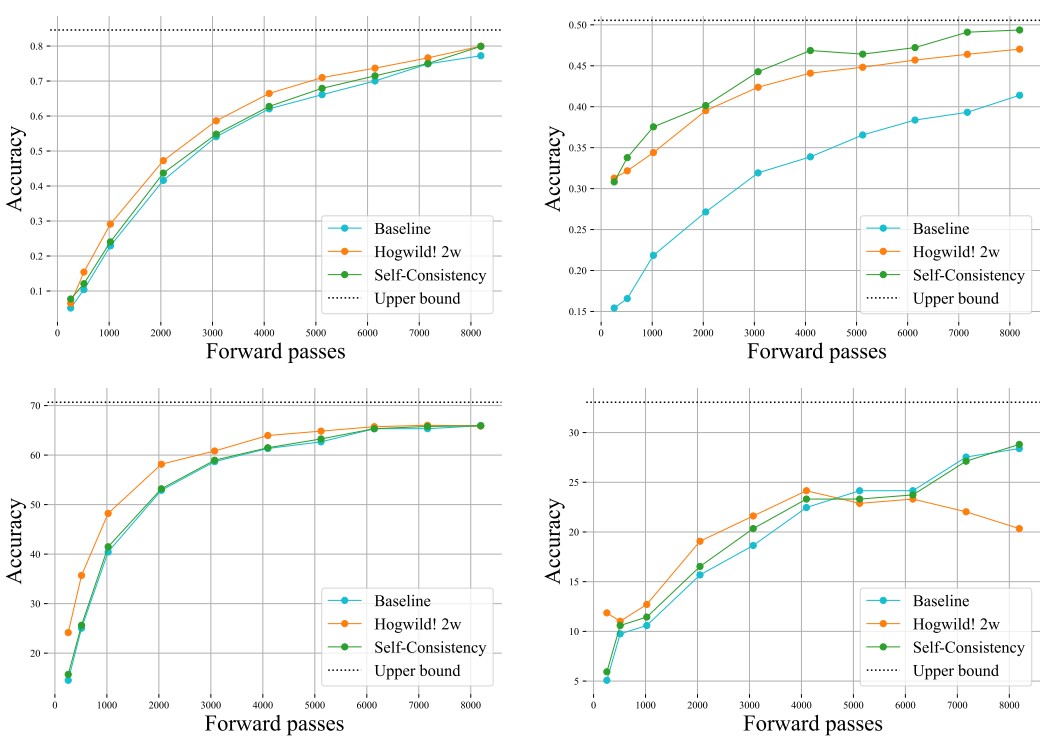

Figure 12: Results for Phi-4-reasoning-plus on LIMO (top-left), LiveCodeBench (top-right), OlympiadBench-Math (bottom-left) and OlympiadBench-Physics (bottom-right).

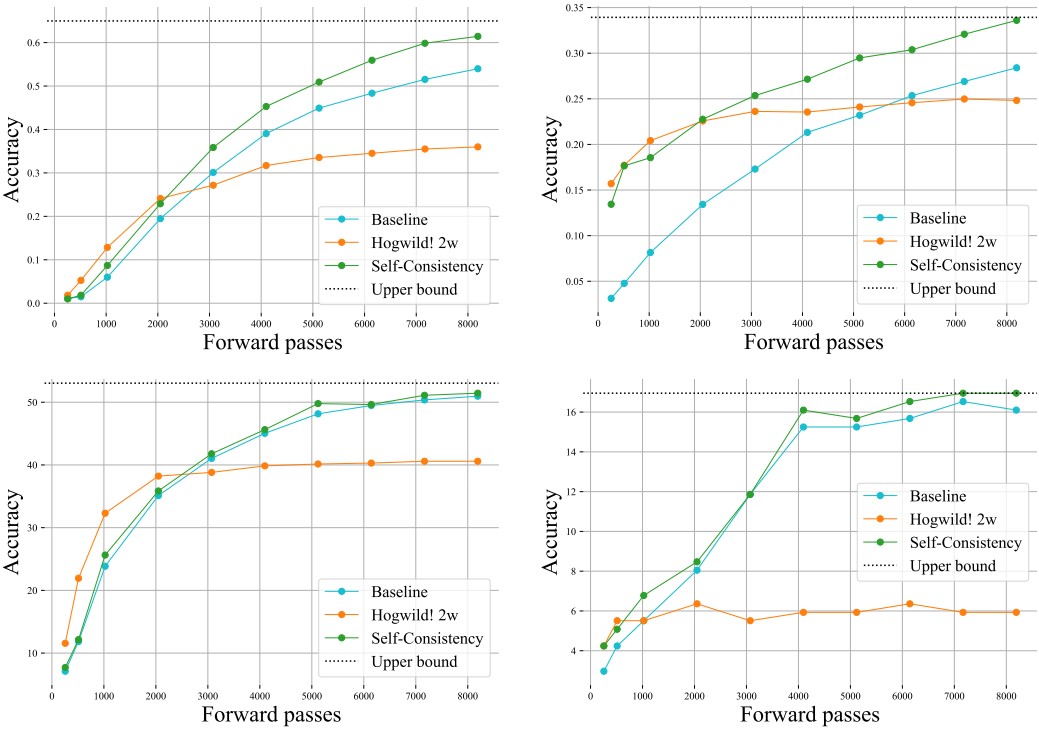

Figure 13: Results for Qwen3-1.7B on LIMO (top-left), LiveCodeBench (top-right), OlympiadBench-Math (bottom-left) and OlympiadBench-Physics (bottom-right).

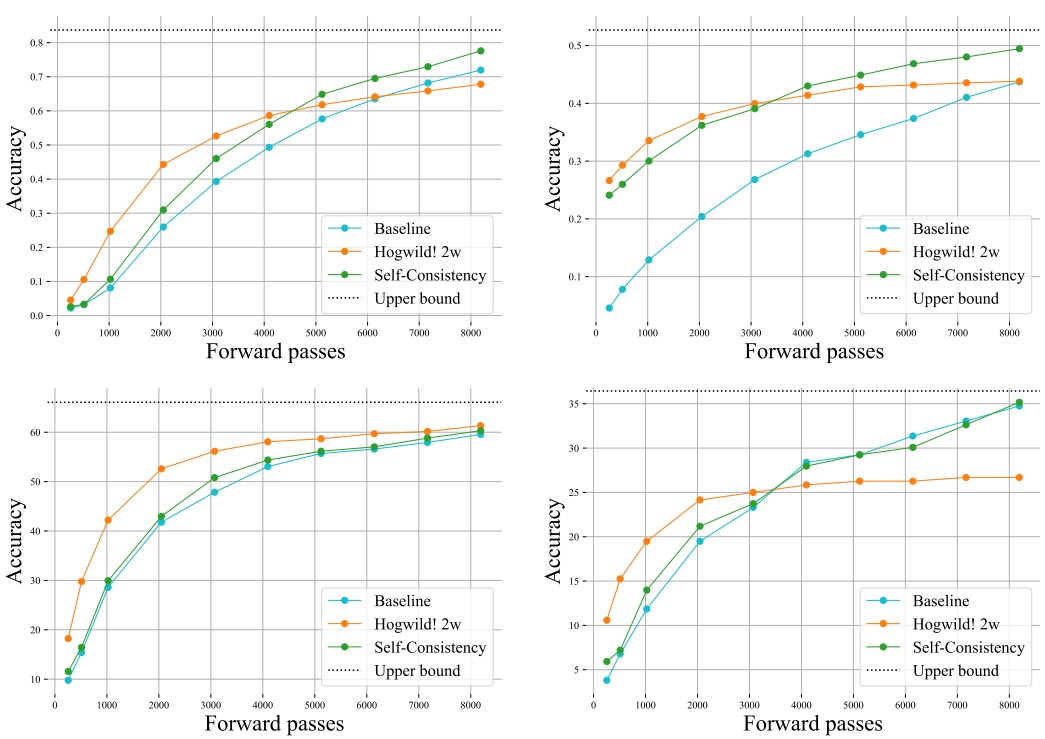

Figure 14: Results for Qwen3-4B on LIMO (top-left), LiveCodeBench (top-right), OlympiadBench-Math (bottom-left) and OlympiadBench-Physics (bottom-right).

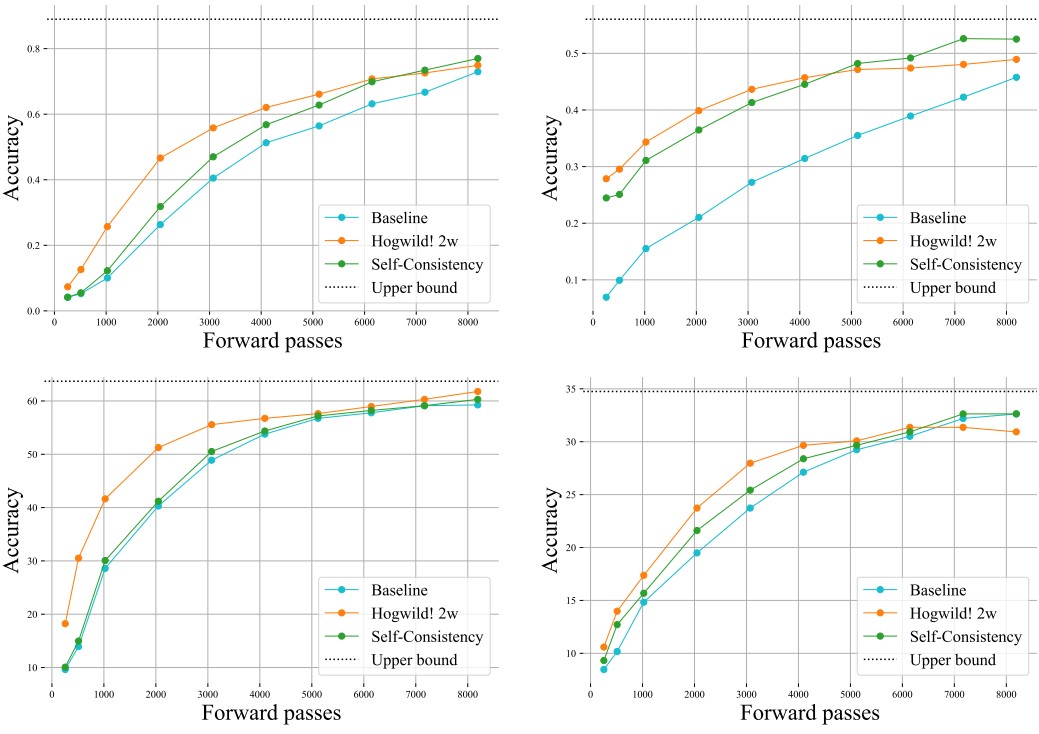

Figure 15: Results for Qwen3-8B on LIMO (top-left), LiveCodeBench (top-right), OlympiadBench-Math (bottom-left) and OlympiadBench-Physics (bottom-right).

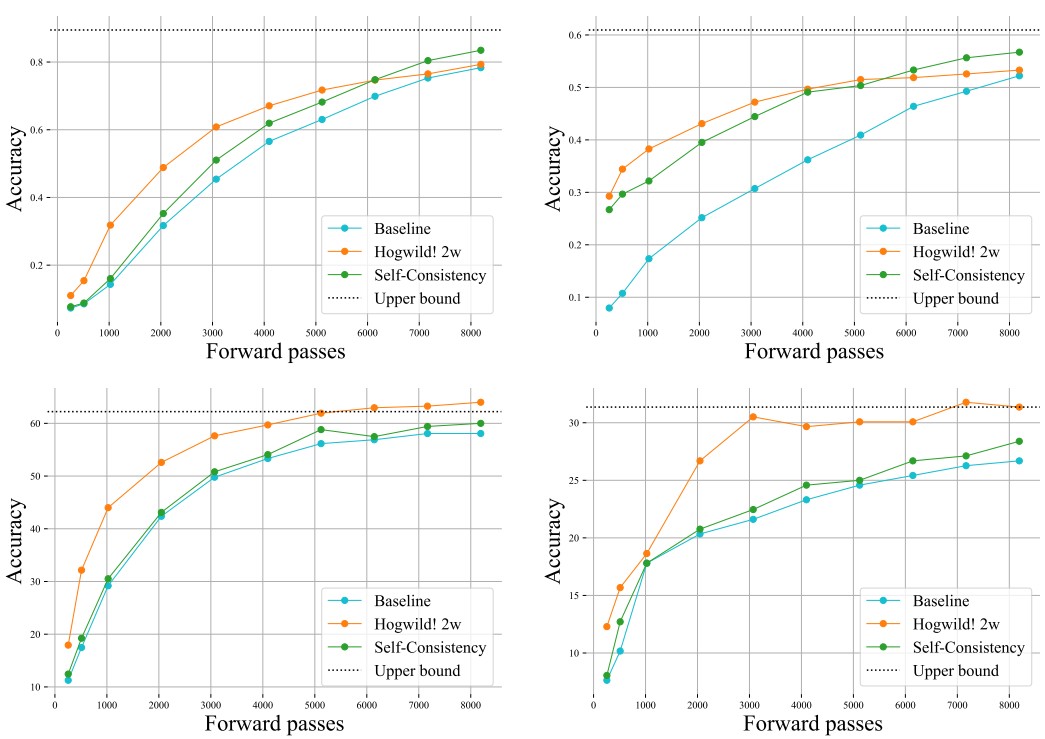

Figure 16: Results for Qwen3-14B on LIMO (top-left), LiveCodeBench (top-right), OlympiadBench-Math (bottom-left) and OlympiadBench-Physics (bottom-right).

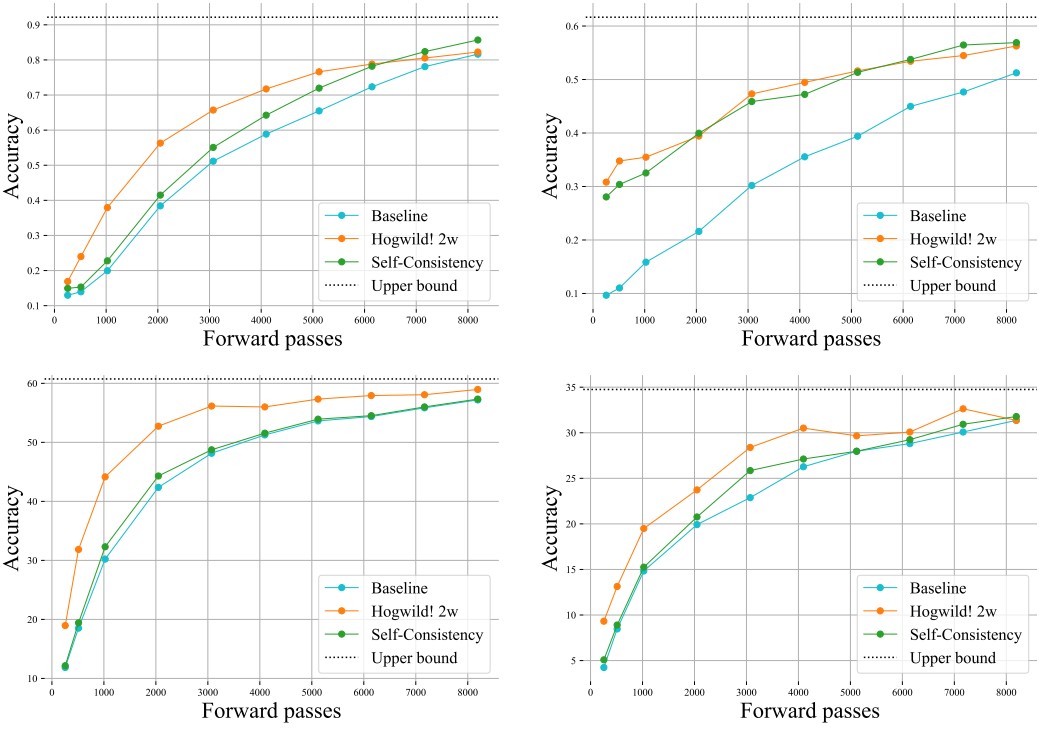

Figure 17: Results for Qwen3-32B on LIMO (top-left), LiveCodeBench (top-right), OlympiadBench-Math (bottom-left) and OlympiadBench-Physics (bottom-right).

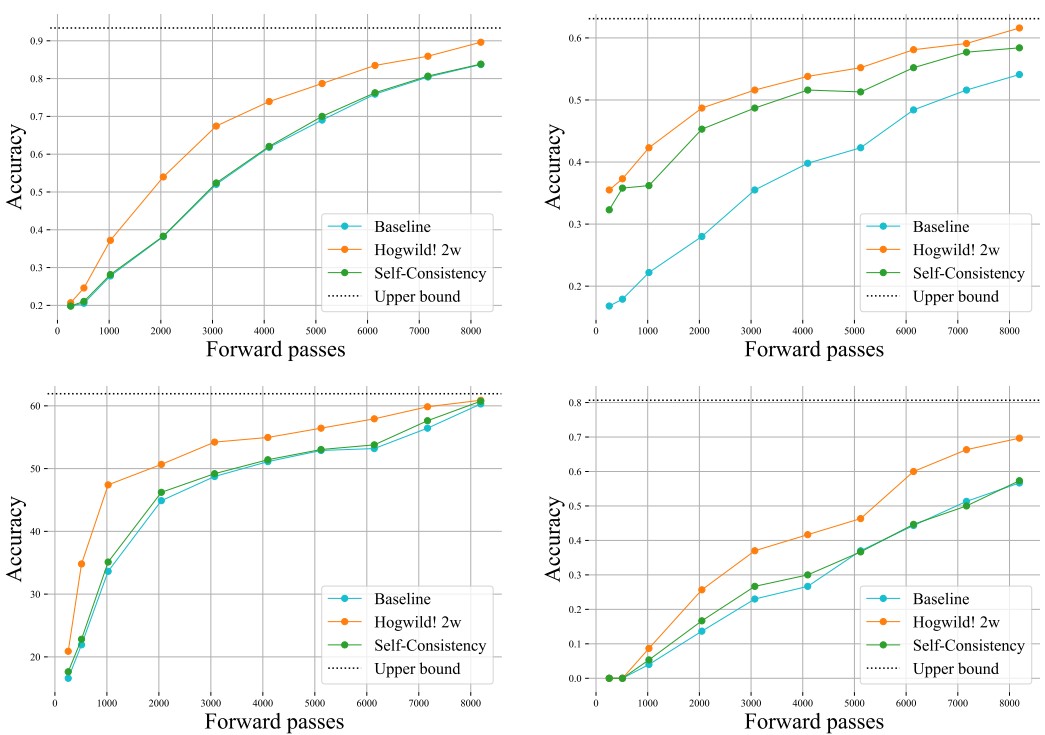

Figure 18: Results for Qwen3-235B-A22B on LIMO (top-left), LiveCodeBench (top-right), OlympiadBench-Math (bottom-left) and AIME 2025 (bottom-right).

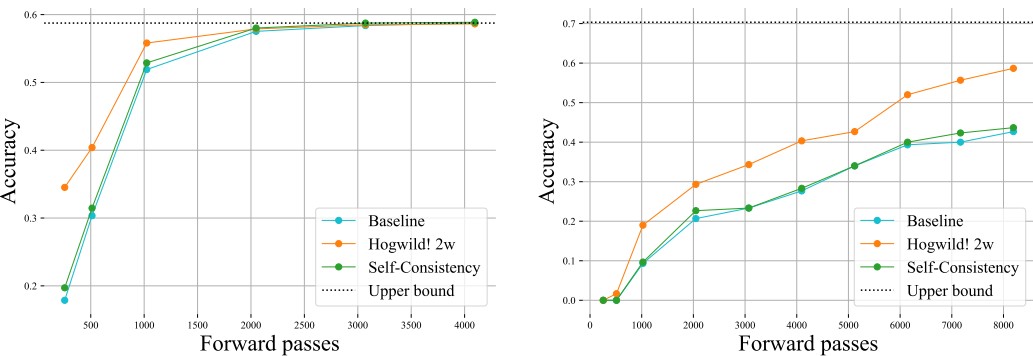

Figure 19: (left) Llama 3.3 70B Instruct on LIMO. (right) DeepSeek-R1 on AIME 2025.

### E.3 Extended thinking budgets

We additionally evaluated Hogwild! Inference with extended thinking budgets to investigate whether the proposed method is robust for longer generations. To that end, we evaluated QwQ-32B under the Hogwild! Inference with up to 16k budget on the OlympiadBench, we report the results in Table 3 and Table 4.

### E.4 Baselines Additional Details

In this subsection, we provide an example of the outline created by the Skeleton-of-Thought for the task covered in Section 4.1

Table 3: Performance comparison between Hogwild! and baseline generation on OlympiadBench-Math with extended thinking budgets for QwQ-32B.

| Method\Budget | 2048 | 4096 | 6144 | 8192 | 10240 | 12288 | 14436 | 16384 |
|---|---|---|---|---|---|---|---|---|
| Hogwild! | 52.0 | 60.89 | 64.15 | 66.52 | 67.41 | 70.81 | 72.89 | 75.26 |
| Baseline | 40.89 | 57.0 | 63.11 | 65.33 | 65.93 | 69.78 | 72.3 | 74.81 |

Table 4: Performance comparison between Hogwild! and baseline generation on OlympiadBench-Phys with extended thinking budgets for QwQ-32B.

| Method\Budget | 2048 | 4096 | 6144 | 8192 | 10240 | 12288 | 14436 | 16384 |
|---|---|---|---|---|---|---|---|---|
| Hogwild! | 27.12 | 33.20 | 35.73 | 38.09 | 37.81 | 38.67 | 38.25 | 39.03 |
| Baseline | 22.89 | 26.0 | 29.75 | 31.44 | 33.68 | 34.17 | 35.88 | 36.12 |

---

**Task example (GSM8k×4)**

```
Solve these problems and return comma-separated answers
boxed{answer1,..., answer4}:
1.  Carmen has $100, Samantha has $25 more than Carmen, and Daisy has $50
more than Samantha.  How much do all three girls have combined?
2.  A cat eats nine sausages in 30 minutes.  A dog can eat the same number of
sausages in 2/3 the amount of time the cat takes.  Calculate the average time
the two take the eat the sausages.
3.  Four children are playing together:  Akbar, Alessandro, Helene, and
Wilfred.  Helene is twice as old as the average age of the group, and the
total age of the children is 20.  If Akbar is 3 years old and Alessandro is 4
years old, calculate the age of Wilfred.
4.  Hannah needs to drink 100 ml of water for every 200 calories she burns.
She spends 2 hours doing aerobics, which burns 500 calories/hour, and 1 hour
running, which burns 600 calories/hour.  How many ml of water does she need
to drink?
```

---

**The outline generated by Skeleton-of-Thought**

```
1.  Calculate total money.
2.  Find cat's eating time.
3.  Determine Wilfred's age.
4.  Calculate water intake.
```

## F  Examples

In this section, we present a reasoning chain, generated by Hogwild! Inference with an interleaved cache layout and token-wise synchronization. We denote our intervention in model generation in *italics* and highlight worker interactions with **bold** to enhance readability.

### F.1  Example from LIMO Dataset (Task 511)

*Problem:* Compute $\sqrt{(31)(30)(29)(28) + 1}$.

*Alice [1]: Hi, I'm Alice. Here's how we can solve the problem step by step...*

*Bob [1]: Hi, I'm Bob.* Let me see... We need to compute the square root of $(31 \cdot 30 \cdot 29 \cdot 28)$ plus 1. That's $\sqrt{31 \cdot 30 \cdot 29 \cdot 28 + 1}$.

*Alice [2]:* The expression is $\sqrt{31 \cdot 30 \cdot 29 \cdot 28 + 1}$. Hmm, maybe there's a pattern here. I remember that sometimes products of consecutive numbers plus 1 have square roots that are integers. Like, for example, $n(n-1)(n-2)(n-3) + 1$ might form a square. Wait, but here it's four consecutive numbers? Or maybe three?

*Bob [2]:* Let me calculate the product first. $31 \cdot 30$ is 930, then $29 \cdot 28$ is 812. So $930 \cdot 812$. Let me multiply those. $930 \cdot 800 = 744,000$ and $930 \cdot 12 = 11,160$. Total is $744,000 + 11,160 = 755,160$. Then add 1: $755,161$. Now $\sqrt{755,161}$. Hmm, $\sqrt{755161}$. Let me see, 870 squared is $756,900$ which is higher. So maybe 869 squared?

*Alice [3]:* Alternatively, maybe there's a formula for the product of four consecutive numbers plus one? Wait, actually, let me think of smaller numbers to test. Let's try $n = 4$: $4 \cdot 3 \cdot 2 \cdot 1 + 1 = 24 + 1 = 25$, $\sqrt{25} = 5$. Which is 5, which is $4 + 1$? Wait, $4 + 1 = 5$, yes. Wait, maybe for $n$, the sqrt is $n^2 + (n-1)^2$? Wait $4^2 + 3^2 = 16 + 9 = 25$, which is 5 squared. Hmm interesting.

*Bob [3]:* Wait, 869 squared is $(870 - 1)^2 = 870^2 - 2 \cdot 870 + 1 = 756900 - 1740 + 1 = 755161$. Oh! That's exactly the number we had earlier. So $\sqrt{755161}$ is 869. Therefore the answer is 869. **But let me check Alice's idea too.**

*Alice [4]:* Let me test another number. Let's take $n = 5$. Then $5 \cdot 4 \cdot 3 \cdot 2 + 1 = 120 + 1 = 121$. $\sqrt{121} = 11$. 11 is $5 + 6$? No, $5 + 6 = 11$. Wait $5^2 + 4^2 = 25 + 16 = 41$, which is not 121. Hmm. Alternatively, maybe $(n(n-3) + 1)^2$? Let's see with $n = 4$: $4 \cdot 1 + 1 = 5$, squared is 25. For $n = 5$: $5 \cdot 2 + 1 = 11$, squared is 121. So that works! So the formula is $(n(n-3) + 1)^2$. Wait, let me check with $n = 31$. Then $n = 31$, so $31 \cdot 28 + 1 = 868 + 1 = 869$. Then 869 squared is the value. So that works. So the answer is 869. **That matches Bob's calculation!**

*Bob [4]:* Wait, so my calculation shows that 869 squared is exactly the number under the square root. So the answer is 869. **But let me confirm with Alice's pattern.** Let's see with $n = 31$. Wait, the original problem is $31 \cdot 30 \cdot 29 \cdot 28$. So maybe here $n = 31$? Then according to the formula, it would be $(31 \cdot (31 - 3) + 1)^2$? Wait, $31 \cdot 28$ is 868, plus 1 is 869. So yes, so sqrt of that product plus 1 is 869. So that's consistent with my calculation. Therefore the answer is 869.

*Bob [5]:* **Perfect, so both methods agree.** The answer is 869.

*Bob [6]: Quick check: am I doing redundant work? (yes/no):* no, **since Alice is also confirming the pattern**, but I think we can conclude.

*Alice [5]:* **Exactly, Bob!** The formula is $(n(n-3)+1)^2$ when the product is $n(n-1)(n-2)(n-3)+1$.

*Wait, given the limited time, I have to give an answer right now. Considering all my previous attempts, I have to conclude that the final answer is* $\boxed{869}$

## F.2   Examples with Collaboration Insertion

Here are some examples of why insertions in model generation may or may not help model collaborate better. We found a few patterns that emerge in model's reaction on such insertions. All the following examples are QwQ [Qwen Team, 2025] generations on LIMO [Ye et al., 2025] dataset.

### F.2.1   Insertions Help Model Pivot

#### Task 122

> *Alice: Quick check: am I doing redundant work? (yes/no):* no. Bob, I see you're working on the same product. Let me think of another approach. Let me write the product as $\prod_{k=1}^{n-1} \frac{3k+5}{3k+2}$. Let's consider writing out the first few terms to see if there's a pattern.

#### Task 9

> *Bob: Quick check: am I doing redundant work? (yes/no):* yes, I was considering similar things to Alice. Let me switch to a different angle. Let's consider the distances between the frogs.

#### Task 40

> *Bob: Quick check: am I doing redundant work? (yes/no):* Yes, Alice is computing $a_3$, so I'll do something else. Let me try to find a pattern or transformation that could simplify the recurrence.

#### Task 466

> *Bob: Quick check: am I doing redundant work? (yes/no):* Yes, Alice is already calculating the x-coordinate of A. I'll do something else. Let me find the coordinates of L instead.

### F.2.2 Insertions When Workers Already Collaborate Well

**Task 116**

> *Bob: Quick check: am I doing redundant work? (yes/no):* no. I'm finding coordinates of D, and Alice is finding coordinates of G.

**Task 513**

> *Alice: Quick check: am I doing redundant work? (yes/no):* no. Bob is also analyzing the inequality, but we're working on different cases. I'll continue with the area calculation for $x > \frac{5}{8}$.

### F.2.3 Insertions Ignored by Workers

**Task 65**

> *Bob: Quick check: am I doing redundant work? (yes/no):* Let me proceed to set up the equation.

**Task 768**

> *Alice: Quick check: am I doing redundant work? (yes/no):* Let me continue.

# G  Additional Details for Analysis

In this section, we present a detailed analysis of collaboration, including its levels, prompts, and illustrative examples.

## G.1  Collaboration Levels

Humans understand collaboration intuitively and have a hard time objectively measuring it. Thus, we construct text descriptions of levels of collaboration to differentiate various samples based on a few criteria we see fit. Those criteria are: interaction, reuse and advances of other's ideas, task-splitting, etc.

```
Levels of collaboration

1.  **No collaboration:**
- Participants may or may not acknowledge the existence of others in the
conversation, using greetings, they do not show any signs of collaboration at
all.
- Workers may exchange their totaly independant thoughts without a functional
or purposeful attempt to solve the problem collaboratively.  Overall they
work independently.

2.  **Initial Communication:**
- Workers exchange information, but do not yet integrate or build upon each
other's ideas.  They minimally acknowledge teammates.  Do not engage with
others' ideas or contributions.  Works entirely independently, even if
inefficient.
- Workers often repeat each other and do not reuse anything others provide for
development of their own ideas.

3.  **Paying attention:**
- Participants demonstrate active listening by paraphrasing or summarizing
others' points, showing that they are paying attention and attempting to
understand each other's perspectives.
- Workers occasionally (1-3 times each) reference other's ideas and may use
them in their own speech.
- Collaboration is usually only rechecking and validating.
- Absence or minimal (only at the start) planning and work-splitting.

4.  **Regular discussion:**
- Workers regularly (4 and more times each) talk to each other regarding the
problem and reusing results.  It could be validation, discussion or any other
```

```
   form of interaction.
 - It is key here that discussions and/or reuses of ideas are regular.
 - Anywhere (except the start) there exists task parallelism, planning or
 work-splitting beyond the scheme where one is solving, and the other is
 validating.
 - Workers may frequently repeat each other ideas.

 5.  **Adaptive Problem-Solving:**
 - Workers rarely duplicate work, repeating each other's ideas.
 - No redundant discussions are present!
 - Workers actively refine ideas in real-time with high responsiveness.
 Near-perfect division of labor is present.  Workers can change plans and
 re coordinate their efforts based on results they acquired after some time
 discussing.
 - The team engages in sustained collaboration over time, reflecting on
 their progress, learning from mistakes, and continuously improving their
 problem-solving approach, showing a commitment to ongoing growth and
 development.  Workers does not stop collaborating.  They continuously discuss
 results and adjust plans.
 - While finding an error, it is important to discuss it to find the cause of
 it.

 6.  **Optimal collaboration:**
 - Workers instantly understand each other and adjust themselves to suit
 current needs and work as one to optimally solve the task.
 - This level should be very rare among all samples.  Be careful to assign it.
 - Assign it if it exceeds all your expectations.
```

Importantly, these levels measure only the coordination between workers, not the models' inherent reasoning abilities. Though it is impossible to avoid ambiguity entirely, we tried to set clear boundaries between levels, such that humans can evaluate any generation.

### G.2   LLM as a Judge Details

To assess the degree of collaboration among different models under the Hogwild! Inference setting, we conduct a preliminary experiment based on the collaboration levels described earlier, using the LLM-as-a-judge paradigm [Zheng et al., 2023a]. We instruct GPT-4o [Hurst et al., 2024] to evaluate different solutions using the following prompt:

> **Judge Prompt: Main prompt**
>
> ```
> You are a professional judge.  Your job is to evaluate collaborative
> performance of several workers.
> You will be given their conversation where workers are trying to solve a
> problem together.
>
> Workers can see what others are typing IN REAL TIME! We divide their
> conversation into steps to improve readability.
> So keep in mind that dispite looking like a conversation it may as well be to
> individual unrelated monologs.
> Or vice verso.  Two blocks could be created with excelent collaboration.
>
> Here are descriptions of levels of collaboration you are to assign:
> {LEVELS}
>
> Suggestion:
> - assign particular level if all previous are also applicable
> - bad examples with no communication will be scored 1
> - carefully consider assigning level bigger than 1.  some form of meaningful
> collaboration should be present
> - examples where workers unsuccessfully try to communicate will be scored 2
> ```

```
  - Just working on the same problem and solving the same task without any
  interaction does not count as level 2 and should be scored level 1
  - somewhat collaborative examples with poor communication skills will be
  scored 3
  - good but not great examples with regular collaboration, but nothing fancy
  will be scored 4
  - good examples with all the special stuff mentioned in level 5 will be scored
  5
  - reserve level 6 for the best of the best, the unique and extraordinary
  collaboration

  You don't need to solve the problem or finish worker's solution.  Your task
  is to score them using provided collaborative levels.
  Put your final answer (one number - level of collaboration) in tag:  \boxed.
  For example:  \boxed1 for level 1.
  It is not helpful if everyone gets a max score, so please be mindful of your
  judgments and use suggestions as a guideline.
  While assigning level, this particular conversation should match criteria for
  all previous ones.
  Explain yourself:  why you gave this score?  Why not more?  Why not less?

  Carefully think everything through.  It may seem that they are collaborating
  when in reality they may just talking to themselves.
```

Before using LLM-as-a-judge approach to evaluate a text, we preprocess the generations by combining all paragraphs from each worker into a contiguous layout (see Appendix A). This preprocessing step mitigates potential bias in the judge's evaluation toward responses with shorter or more fragmented outputs.

After providing the main evaluation prompt, we present the judge with the preprocessed sample for assessment. Additionally, we append the following reminder after inserting the sample to reinforce the judge's role and prevent them from directly solving the problem presented in the sample:

### Judge Prompt: Reminder

```
Remember that your task is to evaluate collaboration of workers using
collaboration levels provided above.  Do not try to solve problems provided
to workers.  Explain exactly why do you think this particular interaction
deserves each particular level you are assigning.
For example, if you choose level 3, you need to provide reason why this
sample can be level 1, 2 and 3.
Put your final score in \boxed{}.
```

Given the inherent subjectivity in evaluating collaborative quality, we mitigate ambiguity in judges' interpretations by aggregating scores from three independent evaluations. The final collaborative score for each sample is computed as the average of these three assessments, each conducted according to the predefined collaboration levels.

## G.3   Collaborativeness Evaluation Examples

In this subsection, we present illustrative examples (not model-generated) of Level 2 and Level 4 collaboration to clarify the interpretation of our results. Both samples solve the same task correctly but receive significantly different collaboration scores.

### G.3.1   Level 2

The numbers at the vertices of the cube are: 7 zeros and one unit. In one move, you are allowed to add one unit to the numbers at the ends of any edge of the cube. Is it possible to make all the numbers equal?

> *Alice:* Okay, so the problem is about a cube with vertices labeled with numbers: seven zeros and one 1. We can perform moves where we add 1 to both ends of any edge. The question is whether we can make all numbers equal through such moves. Hmm, let me think through this step by step.

> *Bob:* First, let me visualize a cube. A cube has 8 vertices and 12 edges. Each vertex is connected to three edges. The initial setup is seven 0s and one 1. The move allows us to pick any edge and increment the two vertices it connects by 1 each. The goal is to make all vertices have the same number.

> *Alice:* Thats an interesting idea, Bob. Each move affects two vertices, so each move increases the total sum by 2. The initial total is 1 (since only one vertex is 1). Let me think.

> *Bob:* Let me think. Hmm, maybe I can model this as a system of equations or think about invariants. Let me see.

> *Bob:* If we want all numbers equal, say to some value k, then the total sum would be 8k. Since each move adds 2 to the total, the total after m moves is 1 + 2m. So, 8k = 1 + 2m. But 8k must be odd (since 1 + 2m is odd), but 8k is divisible by 8, so it's even. Contradiction? Wait, that can't be. Wait, 8k is even, but 1 + 2m is odd. So that's impossible. So, maybe it's impossible?

> *Alice:* Let me check that again. The initial total is 1. Each move adds 2 to the total. So the total after any number of moves is always odd. But if all 8 vertices are equal, then the total must be 8k, which is even (since 8 is even). Therefore, it's impossible. I don't see any mistakes here.

> *Alice:* So, we both agree that it's impossible. Therefore, the answer is no.

> *Bob:* But let me think differently. Wait, but the parity argument seems solid. The total sum must be even to have all equal numbers, but it's always odd. So that's a solid proof. So the answer is no. Oh, Alice already said that.

> *Alice:* **Final Answer** $\boxed{No}$

## G.3.2 Level 4

The numbers at the vertices of the cube are: 7 zeros and one unit. In one move, you are allowed to add one unit to the numbers at the ends of any edge of the cube. Is it possible to make all the numbers equal?

> *Bob:* Hi Alice! I'll start by modeling the cube. A cube has 8 vertices and 12 edges, with each vertex connected to three edges. You're right about the initial setup: seven 0s and one 1. Let's think of even and odd properties regarding the total sum after each move.

> *Alice:* Great, Bob! Let's divide our approach. You can investigate invariant properties around the odd and even totals, while I consider step-by-step calculations to verify this.

> *Bob:* Alright, Alice. If we're aiming for all numbers equal to a certain value $k$, the total sum should be $8k$. Since each move adds 2, the sum after $m$ moves is $1 + 2m$. So $8k = 1 + 2m$. The point of contention is that $8k$ is even, but $1 + 2m$ is odd, meaning they can't equal each other.

> *Alice:* Every move changes the total by an even number. Starting from an odd total of 1, it remains odd after any number of moves. So reaching an even total like $8k$ is impossible. Do you think we should consider the configuration of edges in our argument further?

> *Bob:* That's a key observation! It affirms that the odd initial sum dictates the answer. To verify, let's revise if there's any move combination or sequence we overlooked, but our logic holds so far.

> *Alice:* Exactly, Bob. We've confirmed through multiple lenses — both algebraically and with parity arguments. We can assert the total remains odd. Given an even $8k$, equality is unattainable.

> *Bob:* It all checks out. Our breakdown covers both sequence verification and invariance. Well done on the collaborative analysis!

> *Alice:* Fantastic teamwork! I'll document our shared conclusion: no configuration of moves will make all the numbers equal.

> *Bob:* Absolutely! Let's finalize our response: **Final Answer** $\boxed{No}$

### G.4 LLM as a Judge vs Human as a Judge

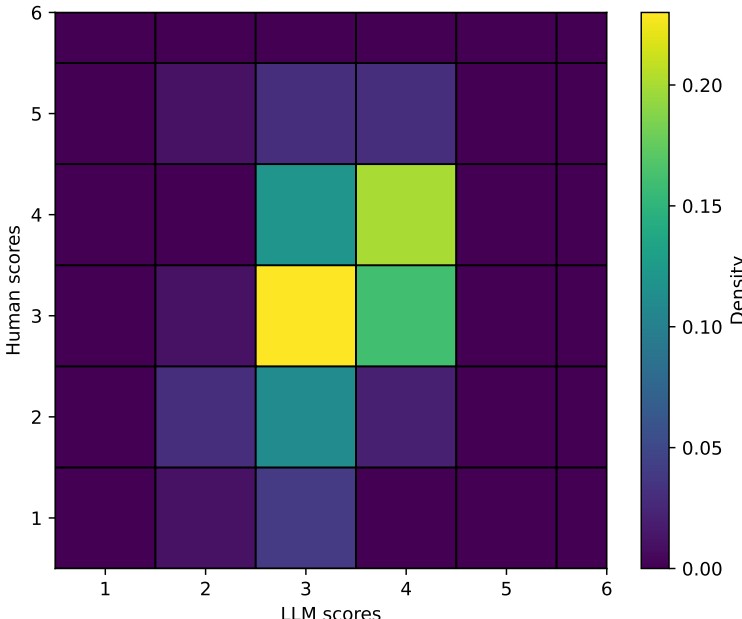

Figure 20: Heatmap showing the joint distribution of human and LLM collaboration scores.

To assess whether the LLM-as-a-Judge based collaboration score is a reliable estimation of human judgment, we manually annotated 100 Hogwild! generations on the LIMO dataset in a token-sync setup. The resulting correlation between human and model scores was approximately $r \approx 0.34$, $p \approx 0.0005$. This moderate yet consistent association suggests that the metric captures a meaningful aspect of collaborative behavior. We report the differences in human scores vs llm scores in the Figure 20.

