# OpenReview forum: "Hogwild! Inference: Parallel LLM Generation via Concurrent Attention"
_NeurIPS.cc/2025/Conference — NeurIPS 2025 spotlight_

### Official Review · Reviewer_JQbb · 2025-07-02

**Clarity:** 2
**Significance:** 3
**Originality:** 3
**Rating:** 5
**Confidence:** 3

**Summary:**

The paper studies a method that has multiple LLM instances to collaboratively generate answers. Instead of pre-defining a framework to decompose a task into sub-tasks and having multiple LLM instances to solve sub-tasks in parallel, the proposed method has multiple LLM instances to figure out the collaboration strategies themselves. The instances collaborate through KV cache. By having different LLM instances to view the same KV cache and update KV cache, they can see each other's progress, potentially avoid redundant works and generate better results.

**Questions:**

What is the cost of this method (how many more tokens are generated?)
In figure 4, 5, 6, when 2 workers are used, does the method use twice as more tokens as the baselines?

**Ethical Concerns:**

["NO or VERY MINOR ethics concerns only"]

**Limitations:**

Yes

**Paper Formatting Concerns:**

No concerns

**Quality:**

3

**Strengths And Weaknesses:**

Strengths:
* Even though the basic idea seems simple, the paper couples multiple interesting ideas to make it work. The ideas are interesting and quite novel.
* The method has been evaluated on multiple reasoning benchmarks (ranging from easy benchmarks and challenging benchmarks) and has been shown good performance improvement on these benchmarks.

Weaknesses:
* The presentation can be improved. For example, fonts and lines in the figures are too small. What is the forward pass?
* The evaluation may not be completely fair. If a forward pass means a token generated by an LLM instance,  Hogwild! Inference with 2 workers should generate twice as many tokens as baseline methods.

---

> ### Author Rebuttal · Authors · 2025-07-31
>
> Thank you for the review and positive feedback! We address the questions below.
>
>  > The presentation can be improved. For example, fonts and lines in the figures are too small.
>
>
> We appreciate the feedback and will improve the presentation clarity, including the figure sizes, in the final version. The additional page allowance post-acceptance will be very helpful for this. Please let us know if you have additional suggestions on improving the presentation.
>
> > (continued) What is the forward pass?
>
> A "forward pass" refers to one invocation of the model to generate the next token or several tokens if inputs are batched. Note that parallel methods, including Hogwild! Inference, Skeleton-of-Thought, Self-Consistency, use batching. We will clarify that in the paper.
>
> > The evaluation may not be completely fair. If a forward pass means a token generated by an LLM instance, Hogwild! Inference with 2 workers should generate twice as many tokens as baseline methods.
>
> You correctly note that token count and forward pass count are not equivalent in Hogwild! Inference: Hogwild! with 2 workers generates approximately twice as many tokens as the baseline for the same number of forward passes. Intuitively, computationally speaking, we get the second token for free!
>
> While a baseline forward pass might be computationally lighter, the actual time-to-solution is highly hardware-dependent and influenced by kernel implementation specifics. Therefore, using forward pass count provides a more hardware-neutral metric for computational effort.
>
> We report token generation speed in Section 4.4 (see Table 1), where Hogwild! inference with 2 and 4 workers can generate tokens nearly 2 and 4 times faster than sequential generation, respectively.
>
> We also agree that it is important to compare the total time needed to solve the problem. To that end, we evaluated the wall inference time vs accuracy using the QwQ-32B AWQ model on the LIMO dataset. **These results can be found in Figure 19, Appendix F.** The results suggest that the benefits of Hogwild! Inference in terms of forward passes translate to faster time-to-solution.
>
> We will move this analysis to the main body (Section 4.4) in the final version of the paper, using the additional content page allowed by this year’s conference rules.
>
> Thank you again for your feedback! If you have any further suggestions or clarification, we are happy to continue during the Reviewer-Author Discussion phase.

---

> > ### Comment · Reviewer_JQbb · 2025-08-02
> >
> > I do not completely agree that the actual time-to-solution is a completely fair metric for comparison. It not only depends on the hardware, but also depends on the batch size. When the inference batch size is small (e.g., 1), twice as much computation generated by two workers is almost completely free in terms of actual time-to-solution. However, when the batch size gets larger, the additional computations may lead to more time to solutions. Therefore, having some additional metrics such as tokens-to-solution can give more full picture of the work.
> >
> > That being said, I agree time-to-solution is the most important metric for evaluating different methods in LLM inference.

---

> > > ### Author Response · Authors · 2025-08-02
> > >
> > > Thank you for suggesting this. There are indeed some caveats to time-to-solution for large batch sizes. Since Howgild! Inference generates one token per worker per step, the number of generated tokens-to-solution with Hogwild! Inference is just the number of forward passes multiplied by the number of workers (plus a few prompt tokens). We will discuss the caveat you mentioned in the final version of the paper, in the inference time section.

---

### Official Review · Reviewer_Qffi · 2025-07-03

**Clarity:** 4
**Significance:** 3
**Originality:** 3
**Rating:** 5
**Confidence:** 2

**Summary:**

The paper introduces “Hogwild! Inference”, a novel parallel inference protocol enabling multiple LLM instances to collaborate dynamically using a shared, concurrently-updated Key-Value (KV) attention cache. This design leverages Rotary Position Embeddings (RoPE) to enable efficient real-time collaboration without predefined strategies. Through extensive empirical evaluation on various reasoning benchmarks (e.g., LIMO, OlympiadBench, LiveCodeBench, AIME’25), the authors demonstrate significant performance improvements, highlighting faster inference and enhanced reasoning accuracy when compared to existing parallel inference strategies.

**Questions:**

Please see the weakness section

**Ethical Concerns:**

["NO or VERY MINOR ethics concerns only"]

**Limitations:**

yes

**Quality:**

3

**Strengths And Weaknesses:**

**Disclaimer:** I am not an expert in this field, but I tried my best to evaluate the work.

**Strengths**

**Novel and Effective Parallelization Strategy**
The approach of dynamically sharing and updating KV caches among parallel inference threads is innovative and practical. It leverages intrinsic LLM reasoning capabilities without additional fine-tuning, representing an impactful advancement in LLM inference techniques.

**Robust Empirical Results:**

Empirical validation is thorough, demonstrating substantial accuracy and inference-time improvements across diverse and challenging benchmarks. Results convincingly illustrate the efficacy and practical potential of the proposed method.

**Clear, Detailed, and High-Quality Presentation:**

I find the paper to be well-written in general.

**Practical Utility and Efficiency Gains:**

The paper includes explicit and detailed evaluations of practical inference-time performance gains, highlighting near-linear scalability in parallel token generation, which strongly supports real-world applicability.

**---XXX---**

**Weaknesses**

**Limited Theoretical Justification for Collaboration Dynamics:**

Although empirically successful, the paper lacks theoretical grounding or deeper insights into why allowing LLM instances to synchronize dynamically via shared caches universally improves reasoning.

**Potential Scalability and Generalization Limitations:**

The method's effectiveness seems dependent on task-specific factors such as task complexity and the intrinsic capability of the LLMs to "self-coordinate." It remains unclear how well the approach generalizes to significantly larger models or vastly different reasoning domains.

**Dependence on LLM’s Intrinsic Collaborative Ability:**

The paper implicitly assumes that LLMs possess sufficient "intrinsic reasoning and collaborative instincts" without clearly specifying or demonstrating the limitations or conditions under which this assumption might fail.

---

> ### Author Rebuttal · Authors · 2025-07-31
>
> Thank you for the constructive review and positive feedback! Let us address the questions below.
>
> > Limited theoretical Jusitification of Collaboration Dynamics
>
> Rigorous theoretical grounding is challenging to achieve, given that modern LLMs are complex, nonlinear systems with sophisticated behavior. We believe the main argument behind the efficacy of Hogwild! Inference is analogous to the rationale for test-time scaling: accuracy on reasoning tasks improves with the number of generated tokens. Hogwild! Inference produces two tokens (or as many as the number of workers) in a single forward pass, yielding a longer chain of thought within the same wall-clock time. The proposed RoPE rotation strategy further enables workers to benefit from each other’s outputs.
>
> > Potential Scalability and Generalization limitations
>
> As shown in Figure 4 (right), the effectiveness of the method depends on the base model’s scale and quality. Smaller models from the same family, such as Qwen3-1.7B, significantly lag behind the single-thread baseline for a fixed number of forward passes, whereas the largest model evaluated (Qwen3-235B-A22B) shows a marked improvement.
> We note that we have validated the efficacy of the proposed method on relatively large models and across various tasks. Specifically, Figure 5 (right) demonstrates experiments with Qwen3-235B-A22B (235B parameters) and DeepSeek-R1 (671B parameters)—the latter being among the largest open-source LLMs with strong performance at the time of submission. Our evaluation setup includes tasks from diverse reasoning domains:
>
> - Math (GSM8k, OlympiadBench-Math, AIME25)
> - Physics (OlympiadBench-Physics)
> - Code (LiveCodeBench)
>
> In addition, we conducted experiments on GPQA-Diamond, which is a highly challenging set of multiple-choice questions in biology, physics, and chemistry. Following the original evaluation protocol, we provide all admissible choices in the question and permute them in a random order to circumvent potential position bias. As observed in the table below, Hogwild! outperforms the baseline for the budgets considered, with the gap being most pronounced for smaller budgets.
>
> ### GPQA-Diamond evaluations (numbered columns are inference budgets)
>
> | Model name  | Method  | 256    | 512    | 1024   | 2048   | 3072   | 4096   | 5120   | 6144   | 7168   | 8192   |
> |-------------|----------|--------|--------|--------|--------|--------|--------|--------|--------|--------|--------|
> | QwQ-32B     | Baseline | 0.2172 | 0.2626 | 0.3535 | 0.4646 | 0.4949 | 0.5707 | 0.5606 | 0.5606 | 0.5505 | 0.5606 |
> | QwQ-32B     | Hogwild! | 0.3939 | 0.4343 | 0.4646 | 0.5404 | 0.5455 | 0.5455 | 0.5758 | 0.5859 | 0.5707 | 0.5657 |
> |||||||||||||
> | Qwen3-8B    | Baseline | 0.2323 | 0.2525 | 0.2879 | 0.3939 | 0.4394 | 0.4747 | 0.5101 | 0.4899 | 0.5000 | 0.5101 |
> | Qwen3-8B    | Hogwild! | 0.3081 | 0.3586 | 0.4394 | 0.5253 | 0.5556 | 0.5505 | 0.5606 | 0.5505 | 0.5505 | 0.5505 |
> |||||||||||||
> | Qwen3-14B   | Baseline | 0.2828 | 0.3131 | 0.3283 | 0.4697 | 0.5404 | 0.5505 | 0.5758 | 0.6162 | 0.5960 | 0.5909 |
> | Qwen3-14B   | Hogwild! | 0.4091 | 0.4192 | 0.5000 | 0.5505 | 0.5657 | 0.5758 | 0.5758 | 0.5859 | 0.5960 | 0.5909 |
>
>
> If you have suggestions regarding additional domains or tasks that could further extend our analysis, we would be glad to consider them.
>
> > Dependence on LLM’s Intrinsic Collaborative Abilitiy
>
> We note that empirical results indicate that the success of multi-threaded task-solving is closely correlated with the model’s capabilities. For small models (e.g., Qwen3 in the 1.7B–8B parameter range, see Figure 4 (right)), Hogwild! does not perform as well for larger budgets. In turn, the benefits become more pronounced for more powerful models that are inherently better capable of collaboration.
>
> We believe that these inherent capabilities are not a fluke, but a direct result of having similar traces in the data the model was trained on.
> As a result, we expect that the reasoning and collaborative abilities of LLMs will continue to improve with further advances in the field.
>
>
> If you have any other questions or concerns, we are happy to discuss them further.

---

### Official Review · Reviewer_HjWC · 2025-07-03

**Clarity:** 3
**Significance:** 3
**Originality:** 3
**Rating:** 5
**Confidence:** 3

**Summary:**

This paper introduces Hogwild! Inference, a method that allows an LLM to explore multiple execution traces for the same problem while enabling synchronization between them. The traces share the attention cache, and the properties of RoPE are exploited to further enhance efficiency.

**Questions:**

How does this method compare with long-CoT-based reasoning? When would be a better scenario to use Hogwild! Inference?

**Ethical Concerns:**

["NO or VERY MINOR ethics concerns only"]

**Final Justification:**

This paper seems to be interesting and solid.

**Limitations:**

yes

**Quality:**

3

**Strengths And Weaknesses:**

Strengths:

1. The idea is very interesting, and the algorithm is supported by both solid math foundations and system design.

2. The authors conduct experiments using large-scale (32B) models such as QwQ, covering a diverse set of tasks including GSM8K, LIMO, LiveCodeBench, and OlympiadBench.

3. They provide illustrative examples, such as the expected collaboration patterns, to help validate their hypothesis.

Weaknesses:

1. The approach is currently for evaluation only. It would be interesting to explore fine-tuning the model toward the desired behavior to evaluate whether further improvements can be achieved.

---

> ### Author Rebuttal · Authors · 2025-07-31
>
> Thank you for your review and positive feedback! We are glad that you appreciate both the idea and experiments & examples. Below, we discuss each of the suggestions raised in the review.
>
> > It would be interesting to explore fine-tuning the model toward the desired behavior to evaluate whether further improvements can be achieved.
>
>
> We note that concurrent and follow-up works already explore various SFT and reinforcement learning strategies for other parallel inference setups [1,2,3,4,5], demonstrating the improvements of the model’s ability to use parallel threads in different setups. For example, [1, 2] describe the SFT + RL pipeline and [4] introduce an SFT dataset for parallel generation. We cite concurrent works in Section 2 and will add a brief summary of follow-up works in the final version of the paper.
>
> While our training-free setup makes Hogwild! inference easier to use, we agree that fine-tuning is an interesting direction for further improvement.
>
> In Appendix B (see supplementary materials, L1032-1045), we discuss how the interleaved cache layout for Hogwild! inference can be computed in parallel during fine-tuning. We plan to investigate methods for improving collaboration between threads, such as fine-tuning and reinforcement learning, in future work.
>
> > How does this method compare with long-CoT-based reasoning? When would be a better scenario to use Hogwild! Inference?
>
> To the best of our understanding, long CoT refers to increasing the generation budget for chain-of-thought to more tokens (e.g., [5,6,7]). Please tell us if you meant something different.
>
> Overall, we believe that Hogwild! Inference with K agents (e.g., 2 or 4) with L step budget can match long CoT inference with *at most K * L budget*, depending on the task. That said, the exact benefit from collaboration does indeed depend on the task. The easiest tasks to parallelize are the ones that have natural independent sub-tasks e.g. see Figure 3 (left) and Section 4.1. More complex tasks such as math/coding problems are harder to parallelize, but still offer significant advantage in terms of faster solutions or greater accuracy given a fixed budget (Section 4.2, figures 11-17 in Appendix).
>
> On the other end of the spectrum, we believe that Hogwild! inference will offer no speed advantage over long CoT in **tasks that are solved with one obvious non-parallelizable algorithm,** such as computing a VDF [8] or similar practical tasks. Furthermore, tasks that require very little chain-of-thought ”effort” (e.g., recalling a single trivia fact) would likely offer no advantage from collaboration.
>
> Note that, for mathematical and programming tasks we evaluate on, long CoT will achieve the same accuracy given greater time (e.g., see “upper bound” in Fig.3-6 for sequential chain of thought with abundant budget). However, Hogwild! Inference offers better time-to-solution due to efficient parallel inference (see Table 1). To better quantify this, we report time-to-solution using the efficient implementation in Figure 19 in Appendix F. We benchmark both Hogwild! and sequential inference on a single NVIDIA L40S GPU in terms of wall clock time and accuracy for a more direct efficiency comparison.
>
> We are be happy to discuss these further and answer any additional questions during the discussion period if needed.
>
>
> [1] in et al., Learning to Keep a Promise: Scaling Language Model Decoding Parallelism with Learned Asynchronous Decoding, 2025
>
> [2] Pan et al., Learning Adaptive Parallel Reasoning with Language Models, 2025
>
> [3] Hsu et al., Group Think: Multiple Concurrent Reasoning Agents Collaborating at Token Level Granularity, 2025
>
> [4] Yang et al., Multiverse: Your Language Models Secretly Decide How to Parallelize and Merge Generation, 2025
>
> [5] Biju et al., SPRINT: Enabling Interleaved Planning and Parallelized Execution in Reasoning Models, 2025
>
> [5] Yeo et al., Demystifying Long Chain-of-Thought Reasoning in LLMs, 2025
>
> [6] Luo et al., Deconstructing Long Chain-of-Thought: A Structured Reasoning Optimization Framework for Long CoT Distillation, 2025
>
> [7] Tang et al., Unlocking General Long Chain-of-Thought Reasoning Capabilities of Large Language Models via Representation Engineering, 2025
>
> [8] Boneh et al., Verifiable Delay Functions, 2018

---

> > ### Comment · Reviewer_HjWC · 2025-08-08
> >
> > Thanks the authors for the rebuttal. I raise my score to accept.

---

### Official Review · Reviewer_wvbH · 2025-07-03

**Clarity:** 3
**Significance:** 3
**Originality:** 4
**Rating:** 5
**Confidence:** 3

**Summary:**

The main contributions are the three-tier cache structure, isolation of page to cache block mapping, and the approach to rotate token queries at an angle rather than all previous keys. The authors demonstrate some evidence that LLMs have the capacity to reason without post-training by collaborating without pre-specified roles or explicit cooperation strategies. The paper is very readable and a joy to read.

**Questions:**

My interpretation of figure 1. Is not what is described as keeping KV memories. Rather than stitch together in different orders, why not use a head node to orchestrate with fewer ops? Other routing-like approaches have shown good results by treating a router as a head node orchestrator.
Where in Appendix B do you discuss ways to scale further? Are you referring to the section on distributed inference?
A more comprehensive discussion of the scalability challenges described in the limitations section would be helpful.
Do you have logs that show the distribution of parallel threads in SoT?

**Ethical Concerns:**

["NO or VERY MINOR ethics concerns only"]

**Final Justification:**

The authors answered all my questions by either pointing out details in the appendices or sharing additional data.
Please see my response to the authors:

On the RL setup, I would like to know if there are clear flaws as to why an RL setup would not work. When you prompt the LLM workers to decide their action, given the latest actions from others, it seems to have a similar problem formulation. Thank you for highlighting that self-consistency and SoT cover voting and roles, and the linkage of Figure 9 in E.1, and the examples in G.2 to support the redundancy claim. The additional examples further the point. The addition of the 16k tokens on the OlympiadBench Math is showing strong convergence between Hogwild and the baseline. What is your interpretation of this? Could you also share the same extension for Physics? There is more plateauing behavior on that benchmark in Figure 5. The 6-worker evaluations in Appendix E.1. and additional LLM-as-judge results on Claude 4 Opus that are comparable to Figure 7. are satisfying. The additional details comparing Figure 9 and Figure 7, and elaborating on collaborativeness vs. output length, strengthen the findings. The point on Sequence Parallelism is fascinating. I look forward to further discussing this after seeing it in the final version. Thank you for directing me to Appendix G for the distribution of parallel threads in SoT. I believe the additional details and context justify an elevated rating.

**Limitations:**

It’s not clear that the approach addresses the limitation of alternative methods that is acknowledged: “... at least some of the agents have to solve the entire problem sequentially, and process (re-encode) each other’s progress. This creates additional computational overhead, which presents challenges for both runtime and memory efficiency Wang et al.”.
The stitching together still grows computation. An analysis should be presented if the authors believe it represents a net improvement.
The evaluation on 2-4 parallel workers does not seem sufficient to generalize results and in particular is below a threshold where communication problems of distributed systems will emerge. The foundations of the LLM-as-a-Judge paradigm are not very convincing, especially for the older GPT-4o model. A more convincing evaluation would be human-evals on this particular task. Authors acknowledge the lack of direct correlation between base accuracy and collaborativeness. Could you show that collaborativeness is not correlated to output length?

**Paper Formatting Concerns:**

No substantial concerns, just write out simple test-time scaling rather than “s1-like”.

**Quality:**

2

**Strengths And Weaknesses:**

The prompting of LLM workers to decide next actions seems analogous to producing states and offloading actions to the LLM in an RL setup. The approach the authors take seems more difficult to generalize. The four cases of “solving parallel sub-tasks, cross-verifying each other, discussing strategy, or pivoting to a new plan” may not have much coverage. The authors also acknowledge the need to tune prompts in 4.2. An analysis of routing strategy between the alternatives that are based on voting, roles, and speed; and showing evidence of these approaches not having better compute and accuracy profiles would make the case stronger. The statement “However, when asked directly, they can spot redundancy and change their approach. Overall, we found that when prompted this way, LLMs often (but not always) detect redundancies in their actions and can determine the optimal course of action.” requires data to justify the claim. In Figure 5. The baselines (dotted lines) for the 32B models have more slope than Hogwild. Given the observation that the authors acknowledge the approach can replace correct answers with mistakes at higher token counts, the results would be more convincing if the forward passes were extended. As a positional paper, the work is novel to the best of my knowledge and significant.

---

> ### Author Rebuttal · Authors · 2025-07-31
>
> Thank you for your feedback and suggestions. We are glad that you appreciate our technical contributions. Below, we discuss the questions raised in the review and address suggestions in the same order as they are written.
>
> > The prompting of LLM workers to decide next actions seems analogous to producing states and offloading actions to the LLM in an RL setup.
>
> While this is not how we originally envisioned Hogwild! Inference, there might indeed be some parallels. If you would like to discuss this further, we can continue during the discussion phase.
>
> > An analysis of routing strategy between the alternatives that are based on voting, roles, and speed; and showing evidence of these approaches not having better compute and accuracy profiles would make the case stronger.
>
> We did compare Hogwild! Inference against **some** of these strategies in Section 4.1, namely voting (self-consistency) and roles (Skeleton-of-Thought has a ‘router’ and ‘workers’), see Figure 3 (left, middle). We agree that further exploration of different alternatives would benefit the paper and will continue this exploration in the revised paper.
>
> > The statement “However, when asked directly, they can spot redundancy and change their approach….” requires data to justify the claim.
>
> In our original submission, we validate this in Appendix E.1 (ablation analysis). In **Figure 9 (left), the “No collab. prompt” (yellow) line** corresponds to a version of Hogwild! Inference *where we do not ask workers to spot redundancies.* As we can see, asking the model to spot redundancies (orange line) can significantly improve its effectiveness. Examples with such behaviour could be found in the Appendix G.2.
>
> To further illustrate this, we provide several new examples of how the model reacts to the collaboration prompt:
>
> Abbreviation: QC = “Quick check: am I doing redundant work? (yes/no)”
>
> ```
> 1)
> Alice [6]: QC:  Let me see Bob is also thinking of the same thing. Wait, maybe we can split tasks. For example, I can list triplets starting with smaller first digits, Bob can do higher ones.
> <..triplets found..>
> Bob [20]: QC:  Let me see Alice is doing {1,3,6} and {1,4,5}? Wait, perhaps we should split the four triplets between us. Let me take the first two and she takes the other two?
>
> 2)
> Bob [7]: QC:  Alice is computing D's coordinates, I can compute E and F instead. Let me switch to that.
> Bob [23]: QC:  Alice is working on angle AEP. I can proceed to compute angle CDP.
> 3)
> Bob [7]: QC:  Wait, Alice just made a good point. If we factor z³, then modulus is |z^3| times |...|. So we can focus on maximizing |1 + 2i - z²|. That's a better approach.
> 4)
> Alice [19]: QC:  Bob is also trying generating functions. Maybe I can try a different approach. Let me consider fixing one variable and counting solutions.
> ```
>
> > In Figure 5. The baselines (dotted lines) for the 32B models have more slope than Hogwild. <…>, the results would be more convincing if the forward passes were extended.
>
> We agree that exploring higher token counts is important. To that end, we evaluated Hogwild! Inference with up to 16k budget on the OlympiadBench Math in the same setup as Figure 5 (left).
>
> |Method\Budget|2048|4096|6144|8192|10240|12288|14436|16384|
> |-|-|-|-|-|-|-|-|-|
> |Hogwild!|52.0|60.89|64.15|66.52|67.41|70.81|72.89|75.26|
> |Baseline|40.89|57.0|63.11|65.33|65.93|69.78|72.3|74.81|
>
> Overall, Hogwild! Inference continues approaching the upper bound quality of the base model as the budget increases. We will include further exploration of longer budgets in the final version of the paper.
>
> > The evaluation on 2-4 parallel workers does not seem sufficient to generalize results and in particular is below a threshold where communication problems of distributed systems will emerge
>
> We evaluate more (6) workers in Appendix E.1. With current models, we found that 2-4 workers are optimal for the tasks we evaluated. Scaling Hogwild! to use even more workers effectively is an interesting direction for future research.
>
> > The foundations of the LLM-as-a-Judge paradigm are not very convincing, *especially for the older GPT-4o model*. A more convincing evaluation would be human-evals on this particular task.
>
> While we agree that LLM as a judge has limitations, it is a popular evaluation strategy for many NLP benchmarks [1,2,3]. To demonstrate that our results are not specific to GPT-4o, we additionally evaluate a newer Claude 4 Opus model from May 22, 2025. We use the same setup as in Figure 7 and compare the three communication types: no communication (voting at the end), synchronizing every paragraph, or every token.
>
> |Method\Model|Qwen3-8B|QwQ-32B|
> |-|-|-|
> |No-sync|1.17|1.24|
> |Step-wise sync|2.08|2.3|
> |Token-wise sync|2.65|3.08|
>
> That said, we agree that human evaluation would further improve the credibility of our analysis and will include the crowdsourced evaluation in the final paper.
>
>
> > Could you show that collaborativeness is not correlated to output length?
>
> In our preliminary experiments, we found that collaborativeness **is** correlated to output length. For instance, in Figure 9 (left), the **“Hogwild! (non-instant)”** line is a version of Hogwild! with less collaborativeness (see Figure 7 ‘Step-wise’). This version solves LIMO tasks systematically slower than full Hogwild! Inference. To control for differences in length, Figure 7 compares only generations with the same output length (using a 4096-token prefix from each worker). We will add these details to the paper.
>
> > Where in Appendix B do you discuss ways to scale further? Are you referring to the section on distributed inference?
>
> We briefly discuss scaling Hogwild! Inference with Tensor, Pipeline, and Sequence Parallelism in L1052-1058. However, we agree that our work would benefit from further discussion of inference scalability. Using Tensor and/or Pipeline parallelism with our approach is no harder than parallelizing traditional batch inference. This is because Hogwild! Inference works independently between attention heads in each layer, so each device can be assigned to process a subset of attention heads and the corresponding KV cache slices.
>
> A more interesting case study for Hogwild! is Sequence Parallelism, where **each device holds a subset of tokens**. This would allow us to reduce cross-device communication, essentially assigning each GPU / node to a subset of workers (e.g., Alice, Bob, etc). The main caveat with this approach is that Hogwild! workers need to attend to each other’s tokens on every inference step.
>
> In distributed inference, this pairs nicely with Sequence Parallelism using Ring Attention [4] or similar. When computing Hogwild! concurrent attention, workers can exchange rotated queries using the All-to-All collective operation (Scatter/Gather) available in most frameworks [5]. After that, each worker computes dot-products between the rotated queries and its local KV cache, and exchanges the partial results as in [4]. Note also that this type of parallelism solves load balancing out of the box since workers generate roughly the same number of tokens. We will further discuss and compare different distributed inference strategies in the final version of the paper.
>
>
> > A more comprehensive discussion of the scalability challenges described in the limitations section would be helpful.
>
> Note that the discussion above only refers to scalability in terms of compute and model size. There is an orthogonal (and equally important) direction of scaling Hogwild! Inference to more workers while maintaining efficient collaboration (e.g., see Fig.9, right in Appendix E.1). This could possibly be achieved with more sophisticated memory layouts or fine-tuning the base LLM to collaborate better with many workers. Both are promising directions for future work.
>
> > Do you have logs that show the distribution of parallel threads in SoT?
>
> We do and we will include these in these examples in appendix G. For GSM8Kx5 (Fig 3, left – 5 independent questions), the SoT ‘router’ works as expected and almost always assigns each thread to a different task, which they subsequently solve.
> As for the more complicated LIMO benchmark, what usually happens is that the initial plan helps to some extent, but fails to account for future developments. Here’s one example:
>
> ```
> # Problem (LIMO task 14)
> Three vertices of a cube are $P=(7,12,10)$ , $Q=(8,8,1)$ , and $R=(11,3,9)$ . What is the surface area of the cube?
>
> # Outline (produced by QwQ-32B as SoT router)
> 1. Find the distance between vertices P and Q.
> 2. Find the distance between vertices Q and R.
> 3. Find the distance between vertices P and R.
> 4. Compare the distances to determine the side length of the cube.
> 5. Calculate the surface area using the side length.
> ```
>
> As you can see, the plan allows for some degree of parallelism. However, the three distances in question are *not* sides of the cube, but its diagonals, and the model learns that as it computes the original angles, but at that point, the outline is already decided. In contrast, workers in Hogwild! Inference does not have a fixed plan and can adjust once they do the initial calculations (e.g., Appendix G.2).
>
> If you have any further questions and suggestions, we are happy to address them during the discussion phase!
>
>
> [1] Zheng et al., Judging LLM-as-a-Judge with MT-Bench and Chatbot Arena MTBench
>
> [2] Zhou et al., Instruction-Following Evaluation for Large Language Models
>
> [3] Li et al., From Crowdsourced Data to High-Quality Benchmarks: Arena-Hard and BenchBuilder Pipeline
>
> [4] Liu et al., Ring Attention with Blockwise Transformers for Near-Infinite Context
>
> [5] Li et al., PyTorch Distributed: Experiences on Accelerating Data Parallel Training

---

> > ### Comment · Reviewer_wvbH · 2025-08-06
> >
> > On the RL setup, I would like to know if there are clear flaws as to why an RL setup would not work. When you prompt the LLM workers to decide their action, given the latest actions from others, it seems to have a similar problem formulation.
> > Thank you for highlighting that self-consistency and SoT cover voting and roles, and the linkage of Figure 9 in E.1, and the examples in G.2 to support the redundancy claim. The additional examples further the point.
> > The addition of the 16k tokens on the OlympiadBench Math is showing strong convergence between Hogwild and the baseline. What is your interpretation of this? Could you also share the same extension for Physics? There is more plateauing behavior on that benchmark in Figure 5.
> > The 6-worker evaluations in Appendix E.1. and additional LLM-as-judge results on Claude 4 Opus that are comparable to Figure 7. are satisfying.
> > The additional details comparing Figure 9 and Figure 7, and elaborating on collaborativeness vs. output length, strengthen the findings.
> > The point on Sequence Parallelism is fascinating. I look forward to further discussing this after seeing it in the final version.
> > Thank you for directing me to Appendix G for the distribution of parallel threads in SoT.
> > I believe the additional details and context justify an elevated rating.

---

### Official Review · Reviewer_BW8Q · 2025-07-03

**Clarity:** 2
**Significance:** 3
**Originality:** 3
**Rating:** 4
**Confidence:** 3

**Summary:**

When solving complex questions, LLMs take a long time. In this study, LLMs were run in parallel to synchronize via a concurrently-updated attention cache, implemented using prompting strategy for these workers to decide how best to collaborate. The evaluation results showed improved performance.

**Questions:**

1) To formulate the problems and solutions formally
2) It would be helpful to compare the proposed model with parallel models (with multiple LLMs/agents)
3) It would be important to evaluate more and larger LLMs (the used models are small with limited performance), especially larger models, considering the Chain-of-Thought designs.
4) It would be important to evaluate the models using more datasets (with complex problems). The model evaluation (better use more and larger models and more datasets) is limited to show the stability and reliability of the strategy.
5) Though the token generation speed is faster, how about the average token generation speed, or the total time needed (combination of a set of workers) for solving the challenging problems.

**Ethical Concerns:**

["NO or VERY MINOR ethics concerns only"]

**Final Justification:**

The review clarified the review concerns.

**Limitations:**

The theoretical innovation is limited.
It would be helpful to compare the proposed model with parallel models (with multiple LLMs/agents)
The model evaluation (better use more and larger models and more datasets) is limited to show the stability and reliability of the strategy.

**Paper Formatting Concerns:**

no concern

**Quality:**

3

**Strengths And Weaknesses:**

Strengths:
Increased token-generation speed
The comparison results showed improved model performance

Weaknesses:
It would be helpful to formulate the problems and solutions formally
It is unclear if it is standard using LLM in parallel or with theory innovation
It would be helpful to compare the proposed model with paralle models (with multiple LLMs/agents)

---

> ### Author Rebuttal · Authors · 2025-07-31
>
> Thank you for the review of our paper! We address your questions and concerns below.
>
> > 1. To formulate the problems and solutions formally
>
> At a high level, the problem we are solving is that of allowing multiple instances of an LLM (“agents”) to read and write tokens concurrently as part of the same “shared memory”. To address this problem, we implement an efficient concurrent version of the KV-cache, at the level of attention kernels.
>
> To evaluate this construction, we use published benchmarks, using their standard problem sets and solution criteria. You can find the exact definitions of problems and solutions in the respective benchmark papers that we cite in Sections 4.1-4.2 and provide dataset links in Appendix D.3.
>
> For example, LiveCodeBench v5 [1] consists of programming problems where the model is prompted to generate a snippet of Python code. Once the generation concludes, the benchmark extracts the final code snippet (denoted with markdown backticks) and runs unit tests that are included in the benchmark. The models are scored on the fraction of coding problems where they produce a valid Python code that passes all unit tests.
>
> > It is unclear if it is standard using LLM in parallel or with theory innovation
>
> As described in Section 2, there are prior methods for LLM parallelism, such as Self-Consistency (voting) and Skeleton-of-Thought (task splitting). However, to the best of our knowledge, SoT has not been previously able to solve the kind of reasoning problems we have in Section 4.2 and on. In contrast, Hogwild! Inference introduces instant cross-worker synchronization and flexible parallelism that allows it to better tackle these more challenging problems (see comparison with SoT in Figure 3, middle on LIMO).
>
> > 2. It would be helpful to compare the proposed model with parallel models (with multiple LLMs/agents)
>
> We compare against several baseline parallel strategies in Section 4.1. The method descriptions can be found in L227-243, and the results are summarized in Figure 3.
>
> Note that these strategies use multiple instances of the same LLM (same as Hogwild! Inference) as opposed to multiple different agents. The reason for this is that multiple instances of the same LLM can be batched (run in the same forward pass) and inferenced more efficiently (see Section 4.3). In contrast, running multiple different LLMs would require loading multiple weight sets and running slower non-batched inference.
>
> To better compare against parallel models, we conduct two additional experiments for the author response: 1) voting (self-consistency) with a larger collective of agents, and 2) voting between different model types. We use QwQ-32B as the base model, QwQ-32B & Phi4-Reasoning-Plus and Gemma3-27b-it for two models, and QwQ-32B & Phi4-R+ & Qwen3-32B for three models. In both cases, the parallel solutions are fed into QwQ-32B which is tasked to aggregate the final response (same as self-consistency in Section 4.1, L244), which is more accurate than simple majority voting.
>
> ### 2 agents (LIMO Accuracy %)
> | Method \ budget | 1024 | 2048 | 3072 | 4096 | 5120 | 6144 | 7168 | 8192 |
> |-|-|-|-|-|-|-|-|-|
> |QwQ x2 voting        |19.02|36.26|50.29|57.27|68.23|74.21|75.47|81.47|
> |QwQ,Phi4R+ voting |20.93|37.45|50.18|58.26|68.79|71.11|76.5|79.80|
> |QwQ x2 Hogwild!    |**26.81**|**45.9**|**59.98**|**67.56**|**73.56**|**77.6**|**80.17**|**82.37**|
>
> ### 3 agents (LIMO Accuracy %)
>
> | Method \ budget | 1024 | 2048 | 3072 | 4096 | 5120 | 6144 | 7168 | 8192 |
> |-|-|-|-|-|-|-|-|-|
> |QwQ x3 voting      |23.75|40.88|55.57|59.49|73.44|75.89|77.97|82.74|
> |QwQ,Phi4R+,Q3-32B voting |25.58|41.86|55.94|60.1|74.66|76.99|78.58|82.86|
> |QwQ x3 Hogwild! |**33.78**|**53.37**|**66.59**|**73.68**|**79.56**|**82.01**|**84.21**|**84.7**|
>
>
> ### 4 agents (LIMO Accuracy %)
> | Method \ budget | 1024 | 2048 | 3072 | 4096 | 5120 | 6144 | 7168 | 8192 |
> |-|-|-|-|-|-|-|-|-|
> |QwQ x4 voting |25.95|43.94|57.53|62.91|75.76|78.21|80.54|83.11|
> |QwQ x4 Hogwild!|**35.74**|**58.26**|**69.52**|**76.87**|**80.66**|**83.23**|**84.33**|**84.94**|
>
>
> The results above show that our approach remains superior to these more standard parallel inference approaches.
>
>
> > 3. It would be important to evaluate more and larger LLMs (the used models are small with limited performance), especially larger models, considering the Chain-of-Thought designs.
>
> Respectfully, we cannot agree with the statement that our experiments are not performed at sufficient scale:
> In the experiments presented in Figure 4 and 6 (right) we evaluated Hogwild! Inference on both large and small LLMs. Our results include evaluations for DeepSeek R1 (671B params) and Qwen3-235B-A22B (235B params). These were **the most capable open-weights reasoning models at the time of the submission** (May 2025) and are comparable to GPT4.x on many of the relevant benchmarks [2,3]. We report 3 more benchmarks with Qwen3-235B-A22B in Figure 17 in Appendix E.2.
>
> > 4. It would be important to evaluate the models using more datasets (with complex problems). The model evaluation (better use more and larger models and more datasets) is limited to show the stability and reliability of the strategy.
>
> We agree that more evaluations are important, but we respectfully disagree that our submission is lacking in that regard. We evaluated on **10 LLMs from 5 model families** (QwQ, Qwen3, DeepSeek-R1, Phi-4-reasoning-plus in the main paper, Llama in Figure 18 in appendix) in **6 evaluation setups** (GSM8Kx5, LIMO, LCB, two main setups from OlympiadBench, AIME). We would be grateful if you could suggest specific additional evaluations that we did not cover.
>
>
>
> In addition, we conducted experiments on GPQA-Diamond, which is a highly challenging set of multiple-choice questions in biology, physics, and chemistry. Following the original evaluation protocol, we provide all admissible choices in the question and permute them in a random order to circumvent potential position bias. As observed in the table below, Hogwild! outperforms the baseline for the budgets considered, with the gap being most pronounced for smaller budgets.
>
>
> ### GPQA-Diamond evaluations (numbered columns are inference budgets)
>
> | Model name  | Method  | 256    | 512    | 1024   | 2048   | 3072   | 4096   | 5120   | 6144   | 7168   | 8192   |
> |-------------|----------|--------|--------|--------|--------|--------|--------|--------|--------|--------|--------|
> | QwQ-32B     | Baseline | 0.2172 | 0.2626 | 0.3535 | 0.4646 | 0.4949 | 0.5707 | 0.5606 | 0.5606 | 0.5505 | 0.5606 |
> | QwQ-32B     | Hogwild! | 0.3939 | 0.4343 | 0.4646 | 0.5404 | 0.5455 | 0.5455 | 0.5758 | 0.5859 | 0.5707 | 0.5657 |
> |||||||||||||
> | Qwen3-8B    | Baseline | 0.2323 | 0.2525 | 0.2879 | 0.3939 | 0.4394 | 0.4747 | 0.5101 | 0.4899 | 0.5000 | 0.5101 |
> | Qwen3-8B    | Hogwild! | 0.3081 | 0.3586 | 0.4394 | 0.5253 | 0.5556 | 0.5505 | 0.5606 | 0.5505 | 0.5505 | 0.5505 |
> |||||||||||||
> | Qwen3-14B   | Baseline | 0.2828 | 0.3131 | 0.3283 | 0.4697 | 0.5404 | 0.5505 | 0.5758 | 0.6162 | 0.5960 | 0.5909 |
> | Qwen3-14B   | Hogwild! | 0.4091 | 0.4192 | 0.5000 | 0.5505 | 0.5657 | 0.5758 | 0.5758 | 0.5859 | 0.5960 | 0.5909 |
>
> If you have suggestions regarding additional domains or tasks that could further extend our analysis, we would be glad to consider them.
>
> > Though the token generation speed is faster, how about the average token generation speed, or the total time needed (combination of a set of workers) for solving the challenging problems.
>
> We report token generation speed in Section 4.4 (see Table 1), where Hogwild! inference with 2 and 4 workers can generate tokens nearly 2 and 4 times faster than sequential generation, respectively.
> We also agree that it is important to compare the total time needed to solve the problem. To that end, we evaluated the wall inference time vs accuracy using the QwQ-32B AWQ model on LIMO dataset. **These results can be found in Figure 19, Appendix F.** The results suggest that the benefits of Hogwild! inference in terms of forward passes translate to faster time-to-solution.
>
> Since this analysis caught your interest, we will move it to the main body (Section 4.4) in the final version of the paper, using the additional content page allowed by this year’s conference rules.
>
> To summarize, we believe that the main concerns raised in the review can be resolved through discussion and the additional results we reported. We respectfully ask the reviewer to reconsider their score based on the additional results. If you have further suggestions or clarifications, please inform us so we can address it in the final version of the paper.
>
>
> [1] LiveCodeBench: Holistic and Contamination Free Evaluation of Large Language Models for Code, Jain et al., 2024.
>
> [2] Qwen3 Technical Report, Qwen Team, 2025
>
> [3] DeepSeek-R1: Incentivizing Reasoning Capability in LLMs via Reinforcement Learning, DeepSeek-AI Team, 2025

---

> > ### Comment · Reviewer_BW8Q · 2025-08-08
> >
> > The rebuttal with updated results explained and addressed most concerns of the review. Along with increasing inference budget, the performance improvement compared with baseline models becomes limited.

---

> > > ### Author Response · Authors · 2025-08-08
> > >
> > > Dear Reviewer BW8Q,
> > >
> > > > The rebuttal with updated results explained and addressed most concerns of the review. Along with increasing inference budget, the performance improvement compared with baseline models becomes limited.
> > >
> > > Thank you for your reply. We would like to further comment on the performance of Hogwild! Inference in higher budgets. Both Hogwild! Inference and baselines rely on the base LLM’s innate ability to reason - represented as ‘upper bound’ in Sections 4.1-4.2. Hogwild! Inference achieves solutions faster due to parallelism (see Appendix F, Fig. 19). However, if the base model cannot solve the problem at all with an infinite budget, we do not expect Hogwild! (or any other training-free method) to solve it either.
> > >
> > > With the increased token budgets (see author response), even the baseline inference can solve the problem well, so there is little room for improvement in accuracy. However, **even with the highest budget, Hogwild! Inference solves the problems significantly faster on average,** because it finds solutions earlier.
> > >
> > > For instance, for QwQ-32B on LIMO benchmark (S4.1) with a budget of 8192 tokens, the improvement in accuracy is ~4% (82.37% vs 78.34%), but **Hogwild! finds the solution on average 21% faster than the baseline due to finding solutions earlier**. Likewise, our GPQA results for the same model (in the response above), QwQ-32B is about 18% faster (on average) at the highest budget. This effect can be seen in more detail by comparing lower budget accuracies in our submission and author response.
> > >
> > > We appreciate that our response explained and addressed most of your concerns. Since the discussion period is nearing its end, we ask you to re-evaluate your score accordingly.

---

### Comment · Area_Chair_2Yc1 · 2025-08-06

Dear Reviewers,

Please review the authors’ rebuttal at your earliest convenience. If you have further questions, use the discussion forum to engage with the authors, and kindly update your review and score as needed.

Thank you for your time and service.

AC

---

### Note · Authors · 2025-08-12

We would like to thank the Reviewers for their feedback and discussion. We are glad that the reviewers appreciate our technical contributions [HjWC, Qffi, JQbb] and strong empirical results [BW8Q, HjWC, Qffi, JQbb] on a diverse set of benchmarks and models [HjWC, Qffi, JQbb]. They also make a number of constructive suggestions, such as additional benchmarks [BW8Q, Qffi], baselines with multiple LLMs [BW8Q], and extended token budgets [wvbH]. The reviewers also asked insightful questions and made suggestions to improve clarity [JQbb, QW8Q, wvbH, HjWC, Qffi]. We will update the paper accordingly. We are glad that the reviewers engaged in constructive discussion and took our responses and additional results into consideration. We would also like to thank Area Chair 2Yc1 for encouraging reviewer-author discussion.

In addition, we would also like to highlight one potentially unfinished discussion with Reviewer BW8Q that took place during the last day of the Reviewer-Author discussion phase, immediately after the reviewer’s response.
Dear Reviewer BW8Q: if our response alleviated your follow-up concern, please consider updating your score based on our discussion. If not, please ask the Area Chair to forward your remaining concerns to us in the verdict so we can understand what they are and address them in a future revision.

---

### Decision · Program_Chairs · 2025-09-17

**Decision:**

Accept (spotlight)

**Comment:**

Hogwild! Inference presents a novel shared KV-cache framework enabling multiple LLM workers to collaborate concurrently without fine-tuning, yielding consistent accuracy gains and reduced latency across reasoning and QA tasks. The approach is simple, general, and well-engineered, integrating cleanly with existing inference pipelines while offering a fresh perspective on collaborative decoding. Experiments span diverse tasks, and the method demonstrates clear benefits over sequential baselines under realistic budgets, with minimal overhead. While broader evaluation and deeper theoretical framing would strengthen the work, the contribution is timely, practical, and of interest to the scalable inference community. I recommend accept.